# Dynamic behavior of the locus coeruleus during arousal-related memory processing in a multi-modal 7T fMRI paradigm

Heidi IL Jacobs[1,2,3†]*, Nikos Priovoulos[2†], Benedikt A Poser[3], Linda HG Pagen[2], Dimo Ivanov[3], Frans RJ Verhey[2], Kâmil Uludağ[4,5]

[1]Gordon Center for Medical Imaging, Department of Radiology, Massachusetts General Hospital/Harvard Medical School, Boston, United States; [2]Faculty of Health, Medicine and Life Sciences, School for Mental Health and Neuroscience, Alzheimer Centre Limburg, Maastricht University, Maastricht, Netherlands; [3]Faculty of Psychology and Neuroscience, Department of Cognitive Neuroscience, Maastricht University, Maastricht, Netherlands; [4]Center for Neuroscience Imaging Research, Institute for Basic Science and Department of Biomedical Engineering, Sungkyunkwan University, Suwon, Republic of Korea; [5]Techna Institute & Koerner Scientist in MR Imaging, University Health Network, Toronto, Canada

**Abstract** A body of animal and human evidence points to the norepinephrine (NE) locus coeruleus (LC) system in modulating memory for arousing experiences, but whether the LC would recast its role along memory stages remains unknown. Sedation precluded examination of LC dynamics during memory processing in animals. Here, we addressed the contribution of the LC during arousal-associated memory processing through a unique combination of dedicated ultra-high-field LC-imaging methods, a well-established emotional memory task, online physiological and saliva alpha-amylase measurements in young adults. Arousal-related LC activation followed amygdala engagement during encoding. During consolidation and recollection, activation transitioned to hippocampal involvement, reflecting learning and model updating. NE-LC activation is dynamic, plays an arousal-controlling role, and is not sufficient but requires interactions with the amygdala to form adaptive memories of emotional experiences. These findings have implications for understanding contributions of LC dysregulation to disruptions in emotional memory formation, observed in psychiatric and neurocognitive disorders.

*For correspondence:
h.jacobs@maastrichtuniversity.nl

†These authors contributed equally to this work

Competing interests: The authors declare that no competing interests exist.

## Introduction

Multiple decades of neuroscience and psychology research uncovered that arousal contributes to enhanced memory performance and that norepinephrine (NE) transmission plays a pivotal role in this phenomenon (*van Stegeren, 2008*; *Roozendaal and McGaugh, 2011*). The major source of NE to the brain is the locus coeruleus (LC), a small structure in the brainstem. The NE-LC system has widespread efferents to almost the entire brain and can modulate cognition, behavior and autonomic tone via its effects on adrenoreceptors in target neurons (*Sara, 2009*; *Sara and Bouret, 2012*; *Samuels and Szabadi, 2008a*).

Research into NE-modulation of emotional memory enhancement has focused largely on the amygdala (*Hermans et al., 2014*; *Phelps and LeDoux, 2005*; *McIntyre et al., 2003*), an important target region of the LC and a critical nexus in memory for arousing experiences. Rodent work consistently demonstrated that blocking β-adrenergic receptors in the basolateral amygdala (BLA) immediately after learning in arousing conditions impairs memory, while infusion of NE or β-adrenoreceptor agonists enhances consolidation and rescinds the impairment, evaluated one to two days later

(*Roozendaal et al., 2008*; *Power et al., 2002*). Correspondingly, stimulation of the LC during the learning phase modulates activity in the BLA and synaptic plasticity in the CA1, CA3 and dentate gyrus (*Lemon et al., 2009*; *Wagatsuma et al., 2018*). While these studies highlight the importance of adrenergic signaling for plasticity and hippocampal-dependent learning, blocking β-adrenoreceptors prior to learning had no effect on subsequent retrieval 24 hr later (*Wagatsuma et al., 2018*; *Khakpour-Taleghani et al., 2008*). It has been suggested that in the context of strong LC stimulation tonic activity can be shifted into phasic activity and other neuromodulators may then elicit similar but possibly weaker plastic processes (*Roozendaal and Hermans, 2017*; *Vazey et al., 2018*; *Hansen and Manahan-Vaughan, 2015*).

In humans, administration of propranolol 90 min before encoding attenuated expected amygdala responses during encoding of emotional stimuli, and even though the propranolol was eliminated at retrieval, hippocampal responses during retrieval of emotional stimuli were absent (*Strange and Dolan, 2004*). In addition, propranolol impaired memory accuracy for emotional events and decreased saliva measures of NE (*Hermans et al., 2014*; *McIntyre et al., 2003*; *Strange and Dolan, 2004*; *Cahill et al., 1994*). These findings suggest, consistent with animal studies, that emotional memory is dependent upon β-adrenergic-related amygdala-hippocampus interactions.

These network interactions can vary over time or be task-dependent. For example, the LC is involved in consolidation, but within a critical time period after learning, the time required for the hippocampus to facilitate the reorganization and stabilization of storing information in the neocortex (*Khakpour-Taleghani et al., 2008*). In humans, increases in NE-tone during encoding, but not during retrieval, modulated memory performance (*Rimmele et al., 2016*). This dynamic behavior stems from the idea that release of NE due to LC-activation modulates, gates and tunes neural activity in a way that optimizes the signal-to-noise ratio (*Sara, 2009*; *Sara and Bouret, 2012*). Thus, NE selectively reinforces brain activity and memory representations only if arousal-related LC activation occurs at the right time and magnitude (*Mather et al., 2016*). When increases in NE are coupled to arousal, they can lead to hippocampal synaptic plasticity (*Lemon et al., 2009*; *Hansen and Manahan-Vaughan, 2015*), and facilitate the dynamic reorganization of neural networks (*Bouret and Sara, 2005*; *Zerbi et al., 2019*). So far, arousal-induced dynamic changes in LC activity and connectivity with its key target regions during task performance has not yet been examined in animals (*Zerbi et al., 2019*).

Motivated by the previous findings, we sought to examine the contribution of the LC directly in vivo in humans to arousal-related memory enhancement and hypothesized that interactions between the LC and MTL regions vary across memory stages, dependent on the level of arousal.

Examining the LC in vivo functionally and obtaining a robust signal in humans is exceedingly difficult given its small size, proneness to noise (pulsatility of surrounding vessels and motion) and its location adjacent to the fourth ventricle. Our recent efforts in ultra-high-field imaging, providing better spatial resolution and signal-to-noise ratio, open up exciting possibilities. Here, we combined 7T-fMRI scanning during an established emotional memory paradigm, our novel 7T-MRI sequence which allows to image the LC at high resolution (*Priovoulos et al., 2018*), with objective measures of arousal, viz. online physiological measures and serial saliva measures, a proxy for NE, in young adults.

## Results

Twenty-seven young right-handed adults (13 female, 20–30 years, mean ± SD = 22.95±1.96) underwent 7T MR imaging which included structural scans (T1-weighted scan, dedicated LC imaging) and ultra-high-resolution blood oxygenation level-dependent (BOLD) signal fMRI. The functional imaging part consisted of a baseline resting-state scan and an emotional memory task, in which participants were required to memorize face-name associations (*Figure 1*). Faces had an emotional (negative) or neutral expression as determined by independent rating. After the encoding phase, a second resting-state scan (termed 'consolidation') was collected. This was followed with a recollection fMRI-scan, during which participants had to indicate whether they recognized the face and, upon endorsement, had to select the name of the face out of three options (all options were seen previously to stimulate recollection instead of recognition processes). During the functional imaging, we collected breathing (respiratory bellows), cardiac information (pulse oximeter) and serial saliva samples for alpha-amylase (sAA) measures, a proxy for catecholaminergic activity. Three fMRI-datasets were

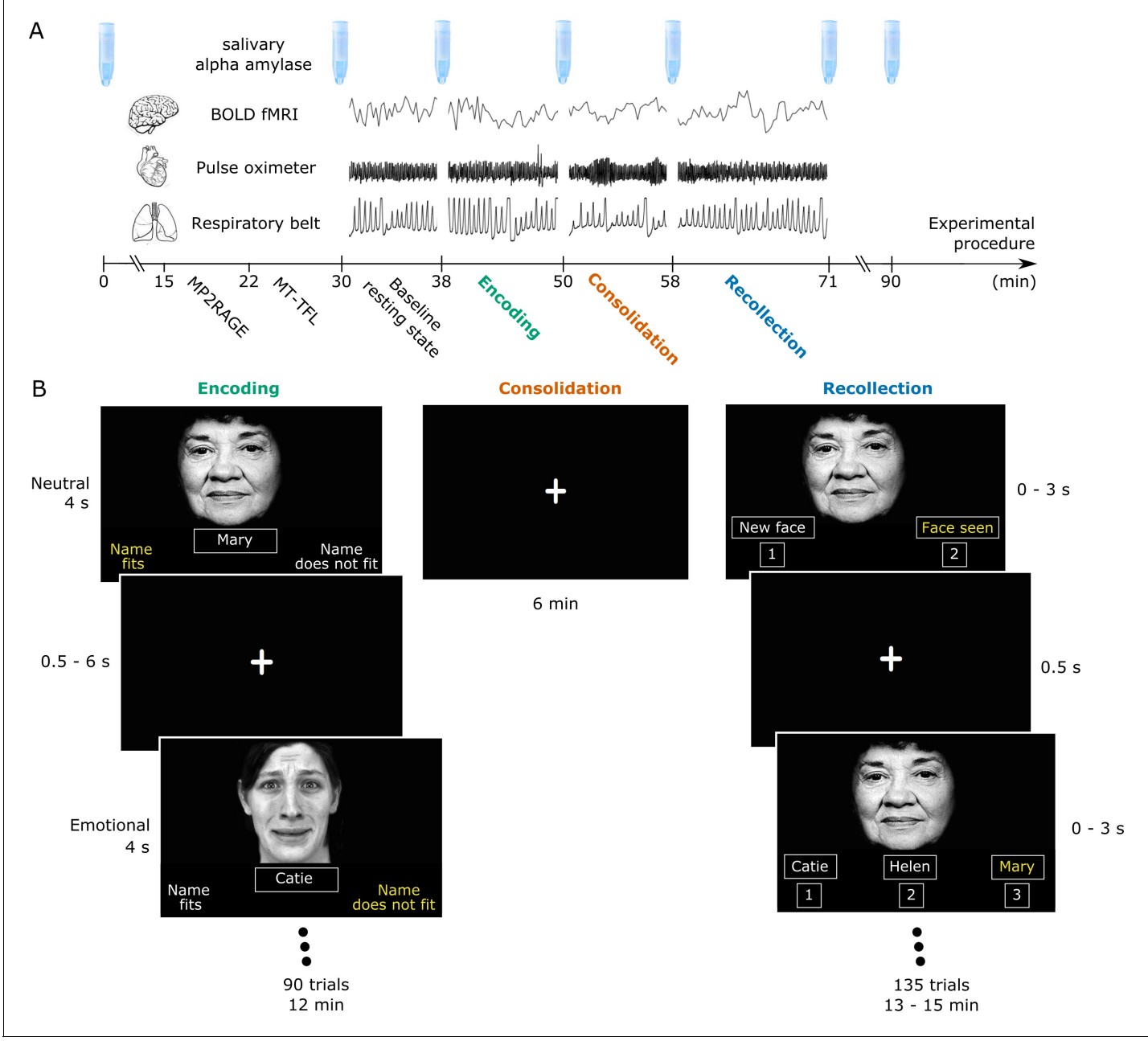

**Figure 1.** Experimental design. (**A**) Task procedure. The salivettes indicate the time points of collection of saliva samples. For the 7T fMRI part, anatomical scans and a baseline 6 min resting state with a fixation cross were collected first. Then a memory task was completed, consisting of encoding, consolidation (a resting-state with a fixation cross) and recollection. (**B**) Encoding consisted of the presentation of 90 face-name pairs (45 emotional- 45 neutral valence; permission of use obtained from PERT96 group). Recollection consisted of 135 trials (45 new faces (23 with emotional valence)) for which participants had to decide if the face was presented before. Upon endorsement, they were asked to indicate the name that was presented with the face during encoding. Throughout the fMRI experiment, breathing and pulse rate data were collected.

The online version of this article includes the following figure supplement(s) for figure 1:

**Figure supplement 1.** Sagittal and axial slices of the T1-weighted and fMRI average in the MNI space at the level of the hippocampus and brainstem.

**Figure supplement 2.** Comparison of the temporal signal-to-noise ratio (tSNR) per participant across fMRI pipelines.

excluded due to technical reasons, one due to spiking because of coil failure one due to a reconstruction error and one due to temporary failure of the response-recording device. *Supplementary file 1* summarizes the neuropsychological test performance of the participants.

## Emotional memory performance is linked to sAA and heart-rate variability changes

We first evaluated the effect of emotional valence on memory performance, as it is well established that emotional information is better remembered than neutral information. All participants performed the task above chance level in recollection (33.3%), with a median number of hits (correctly recollected) of 48.33% (IQR = 40.78, 54.89) and median number of miss rate (incorrectly recollected) of 51.67% (IQR = 45.11, 59.22). There was no significant difference (Wilcoxon paired test: Z = −1.17, p-value=0.244) between the recognition hit rate for emotional (Median = 53.33% (IQR = 45.56, 61.11)) compared to neutral faces (Median = 53.33% (IQR = 44.44, 67.79, *Figure 2A*). However, more false alarms were detected (Wilcoxon paired test: Z = 3.81, p-value<o0.001) for emotional (Median = 22.73% (IQR = 18.19, 34.09) compared to neutral faces (Median = 13.63% (IQR = 9.09, 22.78); *Figure 2B*). We observed no recollection difference (Wilcox test: Z = 0.06, p-value=0.95, *Figure 2D*) between emotional (median = 0.5 (IQR = 0.43, 0.54)) and neutral valence trials (median = 0.45 (IQR = 0.38,0.59), *Table 1*). However, given that performance in the recollection phase may be affected by a response bias in the recognition phase (*Figure 2E–F*), we calculated the false alarm rate and the response bias using the likelihood ratio beta from signal detection theory. A paired t-test revealed a bias in misclassifying emotional faces during recognition (t(27)=5.34,

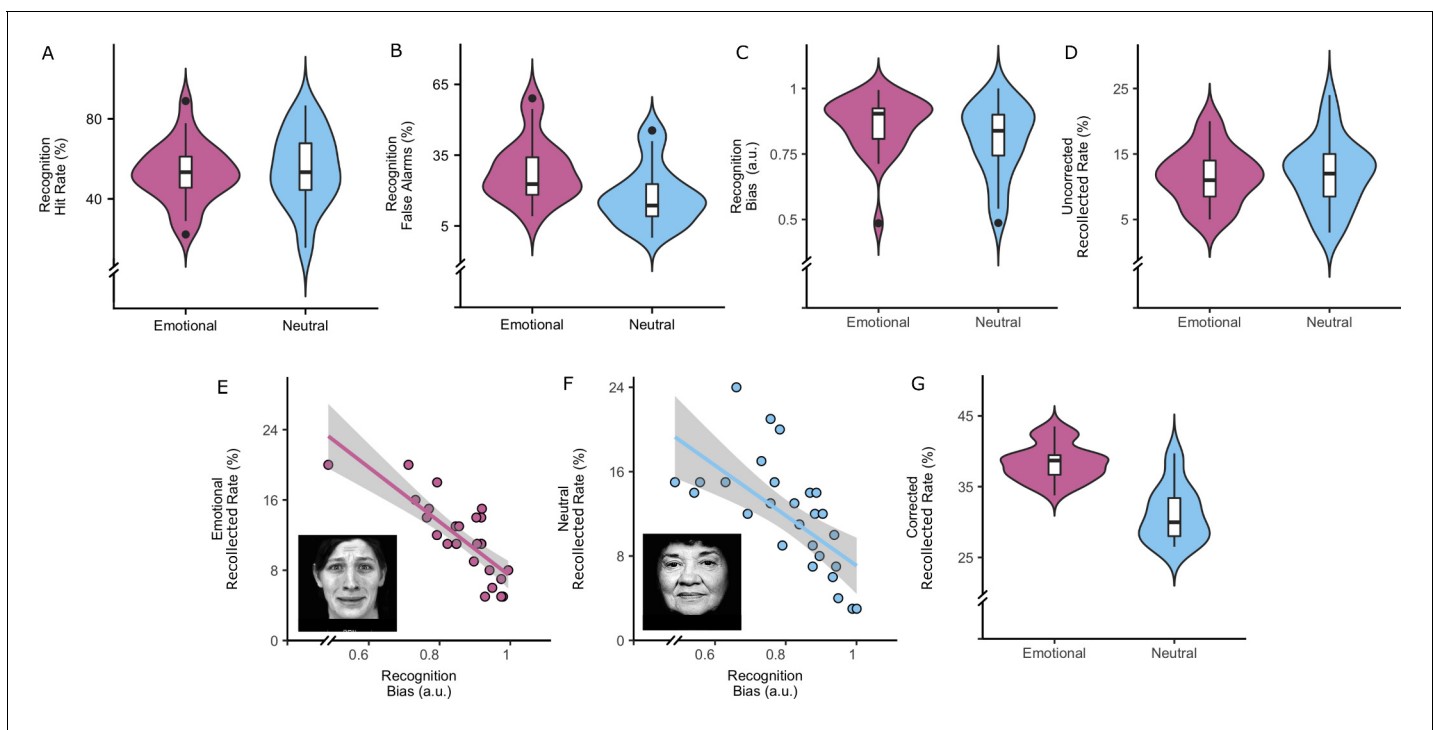

**Figure 2.** Behavioral performance (N = 27). (**A**) Hit rate for emotional and neutral conditions during recognition (paired Wilcoxon test: Z = −1.17, p=0.244). (**B**) False alarm rate for emotional and neutral conditions during recognition (paired Wilcoxon test = 3.81, p<0.001). (**C**) Recognition bias for emotional and neutral conditions (paired t-test, p=0.019). (**D**) Uncorrected recollection rate for emotional and neutral conditions (p=0.95). (**E**) Emotional condition: scatter-plot between recollection rate and bias c (robust linear regression: β = −30.99,t = −9.77, p<0.001). (**F**) Neutral condition: scatter-plot between recollection rate and bias c (robust linear regression: β = −23.78, t = −6.35, p<0.001). (**G**) Corrected recollection rate for emotional and neutral conditions (paired Wilcoxon test: p<0.001). Boxplots inside the violinplots show median, quartiles (boxes) and range (whiskers). Shaded regions depicts the 95% confidence interval.

The online version of this article includes the following source data and figure supplement(s) for figure 2:

**Source data 1.** Behavioral performance on the face-name association task.

**Figure supplement 1.** Schematic representation of the procedure used to calculate bias-free emotional recollection scores.

**Table 1.** Behavioral results of the memory task for emotional and neutral valence.
Note: numbers provided are median and interquartile range (IQR). Differences between emotional and neutral stimuli were tested with a paired Wilcoxon test. All scores, except for the bias express %.

| | Emotional (Median (IQR)) | Neutral (Median (IQR)) | Z | p-value |
|---|---|---|---|---|
| Recognition Hit Rate (Old faces) | 53.33 (45.56, 61.11) | 53.33 (44.44, 67.78) | −1.17 | 0.244 |
| Recognition Miss Rate (Old faces) | 46.67 (38.89, 54.44) | 46.67 (32.22, 55.56) | 1.2 | 0.229 |
| Recognition Correct Rejection Rate (New faces) | 77.27 (65.91, 81.82) | 86.36 (77.27, 90.91) | −3.77 | <0.001 |
| Recognition False Alarm Rate (New faces) | 22.73 (18.18, 34.09) | 13.64 (9.09, 22.73) | 3.81 | <0.001 |
| Recognition Hit Rate - Recognition False Alarm Rate | 23.94 (17.02, 32.83) | 37.68 (28.23, 50.81) | −3.16 | <0.001 |
| Recognition Response Bias | 0.90 (0.81, 0.92) | 0.84 (0.74, 0.9) | 2.29 | 0.02 |
| Recollection Hit Rate Raw | 24.44 (18.89, 31.11) | 26.67 (18.89, 33.33) | 0.35 | 0.726 |
| Recollection Hit Rate (correct recognition) | 50 (43.43, 54.01) | 44.83 (37.97, 58.77) | 0.06 | 0.953 |
| Recollection Hit Rate (Old faces) - Recollection False Alarm Rate (Old faces) | 0.00 (−13.14–8.01) | −10.34 (−24.07–17.54) | 0.06 | 0.953 |
| Bias-corrected Recollection Hit Rate | 38.68 (36.68, 39.46) | 29.97 (28.01, 33.39) | 4.41 | <0.001 |

p-value<0.001, *Figure 2B*) and a higher response bias for the emotional recognition condition (t(27) =2.49, p-value=0.019, *Figure 2C*), towards responding that they had seen the face before. We decided to focus on recollection because of this response bias, our focus on the medial temporal lobe regions and the notion that recollection is more likely to elicit hippocampal activity than recognition (*Rugg and Vilberg, 2013*).

Repeating the analysis after correcting for the bias (partial residuals), showed that participants recollected on average more names with an emotional (median = 38.68, IQR = 36.68, 39.46) relative to those with a neutral valence (median = 29.97, IQR = 28.01, 33.39; paired Wilcoxon test: Z = 4.41, p-value<0.001, *Figure 2G*), in accordance with an expected memory advantage for emotional trials. No reaction time difference between emotional (median = 1.90 (IQR = 1.68,2.10)) and neutral conditions (median = 1.88 (IQR = 1.71, 2.02)) was detected in recollection (Z = 0.09, p-value=0.93) or recognition (Z = −0.43, p-value=0.68; emotional (median = 1.50 (IQR = 1.32, 1.65)); neutral (median = 1.52 (IQR = 1.32, 1.61))). In the remaining analyses, we used this bias-corrected emotional recollection score (*Figure 2—figure supplement 1*; *Table 1* for an overview of all scores). Males and females did no differ in the corrected recollected rate (β = 1.49, z = 1.18, p=0.24), nor did we observe a possible modulation by valence (interaction sex by valence: β = 0.20, z = 0.91, p=0.91).

For task clarification, see *Figure 1*. The 'Recollection Hit Rate Raw' expresses the number of correct recollection of old names divided by the total number of old faces. The 'Recollection Hit Rate (correct recognition)' expresses the recollection success normalized for recognition, that is the number of correct recollection of old names divided by the number of correct recognition of old faces (%Hits). The %False Alarm is 1- %Hits (or Recollection Hit Rate (correct recognition)"). The 'Bias-corrected Recollection Hit Rate' is the bias-corrected recognition rate, as described in *Figure 2—figure supplement 1*. Note that the recollection hit rate for new faces is by definition 0, since they have not been shown in association with a name during encoding.

While there is general agreement in the valence of these images, individual differences in the elicited arousal are common. Therefore, we collected autonomic tone measures. We recorded heart

rate variability (HRV), a proxy of heart rate variability and an index of parasympathetic modulation, particularly in the high (0.15–0.4 Hz) and low-frequency band (0.03–0.15 Hz) (*Hedman et al., 1995*). Root-mean square differences of successive peak-to-peak intervals (rMSSD) were calculated over each time period. RMSSD correlates tightly to parasympathetic activity (*Kleiger et al., 2005*) and is relatively free of respiratory influences. In addition, we measured sAA, a sensitive marker for the sympathetic nervous system, in particular NE.

To facilitate analyzing rMSSD and sAA at similar time points along the paradigm, we calculated the difference in sAA (ΔsAA) levels before and after each of the stages of the fMRI task (resulting in four values per participants, except for missing values due to insufficient saliva; *Supplementary file 2*). Linear mixed effects models with ΔsAA across the task stages revealed an increase in ΔsAA during consolidation relative to encoding (β = −65.95, t(47.92) = −2.91, p=0.033, *Figure 3A*). No differences in ΔsAA were found when comparing the other task stages (*Supplementary file 3A*). These findings did not change when covarying for the baseline sAA measurement outside the scanner or when covarying for sex. To further probe the increase in ΔsAA during consolidation, correlation analyses showed a positive, at trend level, association with emotional memory performance (r = 0.452, N = 18, p-value=0.059, *Figure 3B*).

For the rMSSD, the linear mixed effects models showed no significant change in rMSSD across the task stages (*Figure 3C*, *Supplementary file 3B*). Given that both measures relate to the

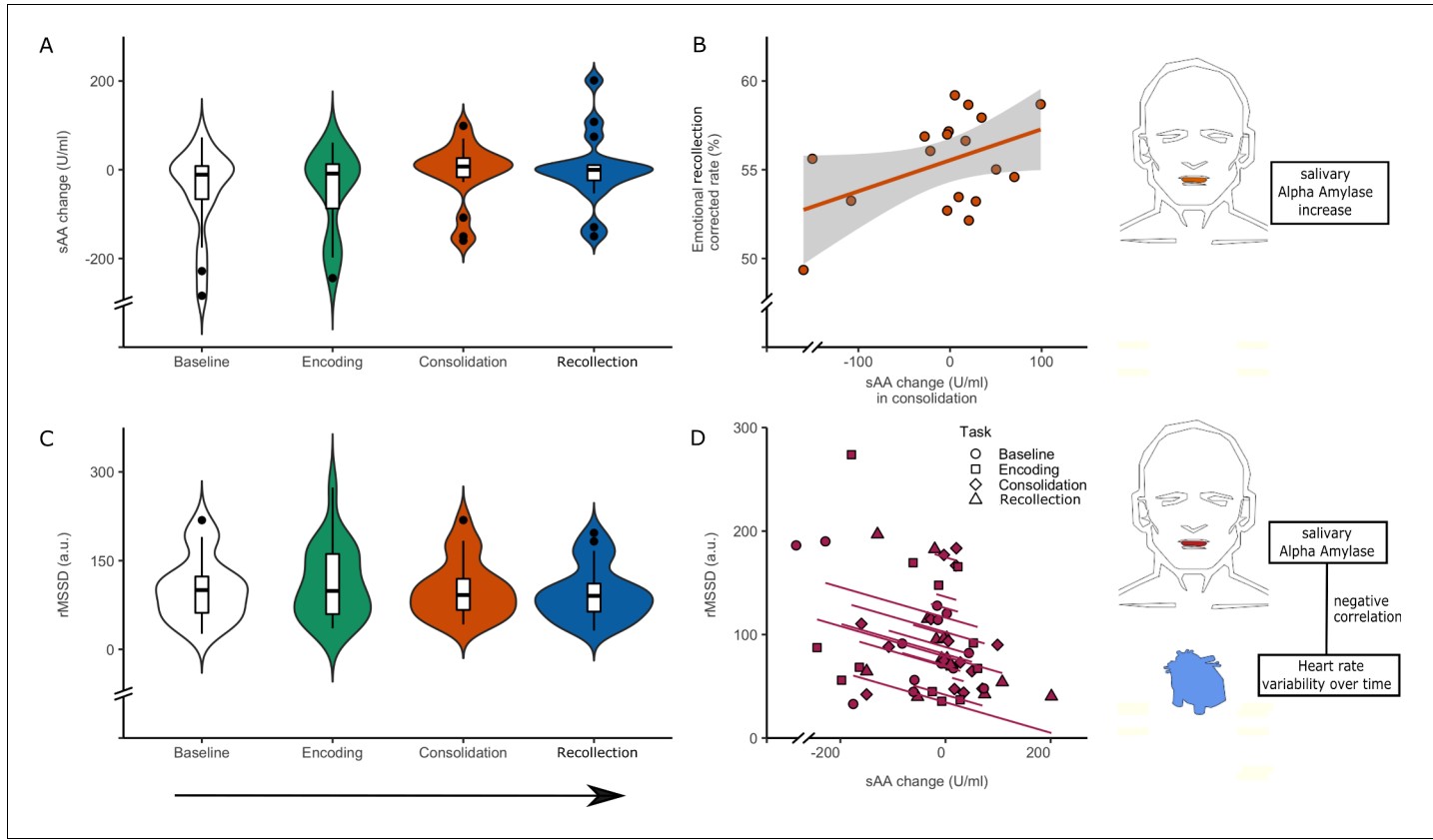

**Figure 3.** Associations between autonomic tone measures and memory performance across the task stages. (**A**) sAA change during the task (encoding, consolidation and recollection) relative to the baseline resting-state. A significant increase in sAA was observed between consolidation and baseline (linear mixed effects model, p=0.033, N = 21, 68 observations). (**B**) Association between ΔsAA (consolidation – baseline) and emotional memory performance (correlation, r = 0.452; p=0.059, N = 18). (**C**) Distributions of the rMSSD during the task (encoding, consolidation and recollection) relative to the baseline resting-state (no significant differences, linear mixed effects model, p=0.23, N = 18, 63 observations). (**D**) Repeated measures correlation plot between ΔsAA and rMSSD across all task stages (R_rm = −0.387; p=0.018, N = 17, 56 observations). Boxplots inside the violinplots show median, quartiles (boxes) and range (whiskers). Shaded region in the scatterplot depicts the 95% confidence interval.

The online version of this article includes the following source data for figure 3:

**Source data 1.** Autonomic tone and memory performance across the entire paradigm.

autonomic system, we then correlated ΔsAA to RMSSD across the stages using bootstrapped (n = 5000) repeated measures correlations and observed a negative relationship ($R_{rm}(df)=-0.387$, (35), p=0.018, CI (95%)=[$-0.638,-0.061$], *Figure 3D*). These observations suggest that perturbations of arousal with a NE-dominant response, reflected in increases in sympathetic and decreases in parasympathetic tone, can enhance memory performance in healthy adults.

## LC activity couples with autonomic tone across the task stages

The LC is the centerpiece of the brain's sympathetic noradrenergic system, and is known to control autonomic function. Therefore, we examined the relationship between LC fMRI and HRV timeseries across the task stages.

Using brainstem-specific group-level independent-component-analyses (ICA) on the baseline resting-state fMRI in combination with a custom anatomical LC template from our MT-TFL sequence, we identified a bilateral LC component (*Figure 4A and B*). To assess the specificity of our findings to the LC, we also investigated a control component in the pons (reference ROI). In addition, sensitivity analyses were done by evaluating the outcomes across four different preprocessing pipelines, each time adding in more physiological noise correction (*Figure 4—figure supplement 1*).

Linear mixed effects models showed no differences across task stages for LC BOLD variability (*Supplementary file 3C*). Consistent with our previous analyses, we applied repeated measures correlation and observed a negative correlation between LC variance and rMSSD across task stages ($R_{rm}(df)=-0.326,(63)$, p-value=0.008, CI (95%)=[$-0.531,-0.084$], *Figure 4C*). We found no correlation between rMSSD and variance in a control region, a component close to the 4th ventricle that explained a similar amount of variance in the ICA ($R_{rm}(df)=-0.201(63)$, p=0.11, CI (95%)=[$-0.494$, 0.427], *Figure 4D*). We also observed no correlation between LC variance and ΔsAA ($R_{rm}(df)=-0.006,(38)$, p-value=0.971, CI (95%)=[$-0.325$, 0.314], *Figure 4E*) or between the reference component variance and ΔsAA ($R_{rm}(df)=-0.012,(38)$, p-value=0.943, CI (95%)=[$-0.309$, 0.330], *Figure 4F*). These findings were reproduced across preprocessing pipelines indicating the robustness of our results (*Figure 4—figure supplement 2*).

In the previous analyses, we demonstrated a negative correlation between LC variability and rMSSD or ΔsAA across the entire paradigm. These results echo previous work demonstrating that blocking β-adrenoreceptor increases HRV (*Bittiner and Smith, 1986*). However, rMSSD reflects beat-to-beat variance in HR in the time-domain and is relatively non-specific to the frequency-domain. Autonomic fluctuations due to emotionally salient stimuli can alter the power of frequencies in the HRV and this can occur in concert with changes in LC BOLD responses (*Bittiner and Smith, 1986*; *Akselrod et al., 1981*; *Lane et al., 2009*). Therefore, we applied spectral analyses to assess the nature of autonomic modulations on LC activity at each task stage by calculating the coherence magnitude squared between the timeseries of the HRV and LC for each task stage (see example in *Figure 5A*). By using a within-subject design with a baseline measure we can eliminate interindividual differences in blood pressure, respiration and, by combining HRV fluctuations with other measures related to the autonomic system, we are well-positioned to differentiate autonomic contributions to LC activity due to emotion or arousal across the different memory stages. Here, we will use the FIXed+explicit Phys pipeline to remove respiratory confounding.

We ran linear mixed effect models to examine whether the relationship between frequency and coherence varied across task stages (*Supplementary file 7A*), followed by simple slopes analyses to determine the region of significant (RoS) moderation at α <0.05. Greater LC-HRV coherence levels were observed during baseline compared to consolidation (β = −0.20, t(7889) = −6.65, p<0.001, RoS:<0.27 Hz), during encoding compared to consolidation (β = 0.13, t(7889) = 4.31, p<0.001, RoS: <0.29 Hz), during recollection compared to baseline (β = −0.20, t(7889) = −6.55, p<0.001, RoS: >0.25 Hz) and during recollection compared to encoding (β = −0.12, t(7889) = −4.17, p<0.001, RoS:>0.27 Hz; *Figure 5C*).

For the control region (*Figure 5B and D*, *Supplementary file 7B*), coherence between the reference region and HRV coherence was greater during recollection compared to baseline (β = −0.13, t (7889) = −4.15, p=0.001, RoS:>0.28 Hz), or at trend-level compared to consolidation (β = −0.09, t (7889) = −2.95, p=0.050, RoS:>0.33 Hz). These findings were replicated across preprocessing pipelines, except for the spatially normalized pipeline where associations between frequency and coherence were noisier (*Figure 5—figure supplements 1–2*, *Supplementary file 4–5*). These patterns indicate greater LC activity coupled to greater parasympathetic inhibition during baseline and

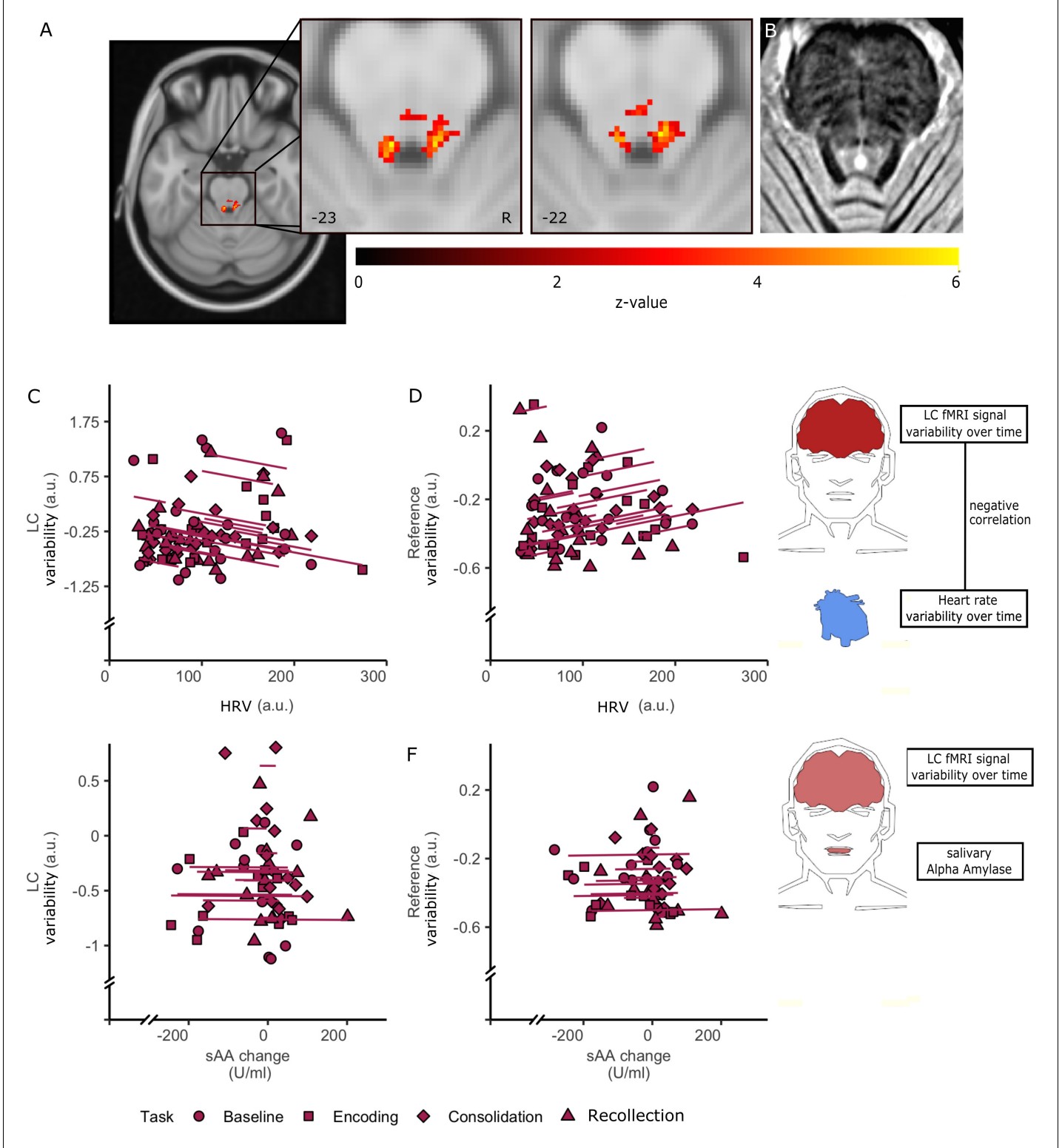

**Figure 4.** Associations between LC BOLD and autonomic tone measures using repeated measures correlations. (**A–B**) LC component visualization. (**A**) Group average t-map from the dual-regression showing a robust bilateral LC component, thresholded at p<0.01 FWE, after denoising with FIX (N = 24). (**B**) LC template from all scans. A hyperintensity can be observed close to the 4th ventricle. (**C–F**) Repeated correlations between LC variability rMSSD (p=0.008, N = 23, 92 observations) or ΔsAA (not significant, p=0.971, N = 17, 56 observations) (**C,E**). Similar associations are shown for variability

*Figure 4 continued on next page*

*Figure 4 continued*

in the reference component (**D,F**) (no significant relationships, p=0.109 and p=0.942, respectively, N = 17, 56 observations). Each marker signifies a participant and distinct shapes show the different task stages.

The online version of this article includes the following source data and figure supplement(s) for figure 4:

**Source data 1.** Locus coeruleus BOLD and autonomic tone measures across the entire paradigm.
**Figure supplement 1.** Functional template of the locus coeruleus during resting-state.
**Figure supplement 2.** Relationship between LC variability and rMSSD across the task stages for each preprocessing pipeline.

recollection, which has been associated with better discrimination between emotional versus neutral faces, attentional resources and better retrieval of context information from stored representation to perform memory task successfully (*Lane et al., 2009*; *Thayer et al., 2012*). On the other hand, consolidation demonstrated lower coherence compared to encoding or baseline, which is consistent with the premise that lower HF-HRV is associated with higher arousal. The latter is consistent with the increase in ΔsAA during consolidation (*Figure 3B*).

To further probe the importance of coupling between autonomic tone and LC variability, we defined the frequency band of maximum coherence as the confidence interval of the median of the maximum-individual coherence between LC and HRV (median = 0.217, 95% CI=[0.189, 0.317]. Interestingly, the width of this CI also nicely fits with the standard definition of high-frequency band of parasympathetic influence on HRV [0.15, 0.4 Hz]). The median coherence within this band was extracted for each individual across conditions to obtain a single point-estimate for our repeated measures correlation analyses with ΔsAA. We observed a negative relationship between LC-HRV median coherence and ΔsAA ($R_{rm}$(df)=−0.383 (34), p-value=0.021, CI (95%)=[−0.64,–0.05], *Figure 5E*). We replicated this analysis across preprocessing stages. Additionally, we repeated this analysis for the reference component, but observed no significant relationship with ΔsAA ($R_{rm}$(df) =0.218 (34), p-value=0.201, CI (95%)=[−0.13, 0.52], *Figure 5F*). Finally, no significant correlation was detected between LC-HRV coherence during any of the stages and emotional recollection rate (all ps > 0.05). The covariation between LC activity and HRV demonstrate that even though both emotion and cognition operate simultaneously, encoding and recollection are coupled to moment-to-moment autonomic regulation supporting cognitive functions required for the task at hand. However, this LC-parasympathetic coupling, appertaining to the central autonomic network, is not sufficient to predict accurate emotional memory recollection. Conversely, LC BOLD signal during consolidation may be specifically tied to arousal-related sympathetic tone, an inference supported by our ΔsAA analyses and the at-trend correlation between ΔsAA during consolidation and memory performance. We should note that the physiological relevance of high-frequency fMRI signal remains unclear, but it has indeed been linked to rapid modulation of the brainstem driving information exchange (*Billings et al., 2018*).

## Arousal-related successful encoding elicits LC activation

We hypothesized that the LC and amygdala would be activated during encoding of emotional compared to neutral faces. The GLM within our predefined mask (*Figure 6—figure supplement 1*) largely confirmed this, by showing activation in the CA1 and 3, bilateral BLA (extending into the centromedial amygdala) and right entorhinal cortex during successful encoding versus no successful encoding (*Supplementary file 8A*, *Figure 6A*). However, no significant clusters were detected for emotional >neutral or for emotional successful encoding >neutral successful encoding.

Given our evidence of the LC's role in modulation of the autonomic system, we examined contrasts from HRV-convolved clusters. No significant clusters were observed after cluster correction in the MTL or pons. But given the limited signal-to-noise ratio typical for brainstem responses and the small-elongated shape of the LC (10–12 voxels in our high-resolution images), we performed our analyses at an uncorrected p<0.05 within a small-volume search space. The high arousal (local HRV minima) events (High Arousal > Low Arousal) showed significant bilateral activation at the LC (peak z-value = 3.3, *Figure 6C*), consistent with our previous results examining HRV-LC coherence during encoding. In addition, significant voxels of LC activation were observed for the interaction of high compared to low arousal during successful encoding (High Arousal successful Encoding > Low Arousal successful Encoding; peak z-value = 5.2; *Figure 6B*), successfully encoding versus forgotten

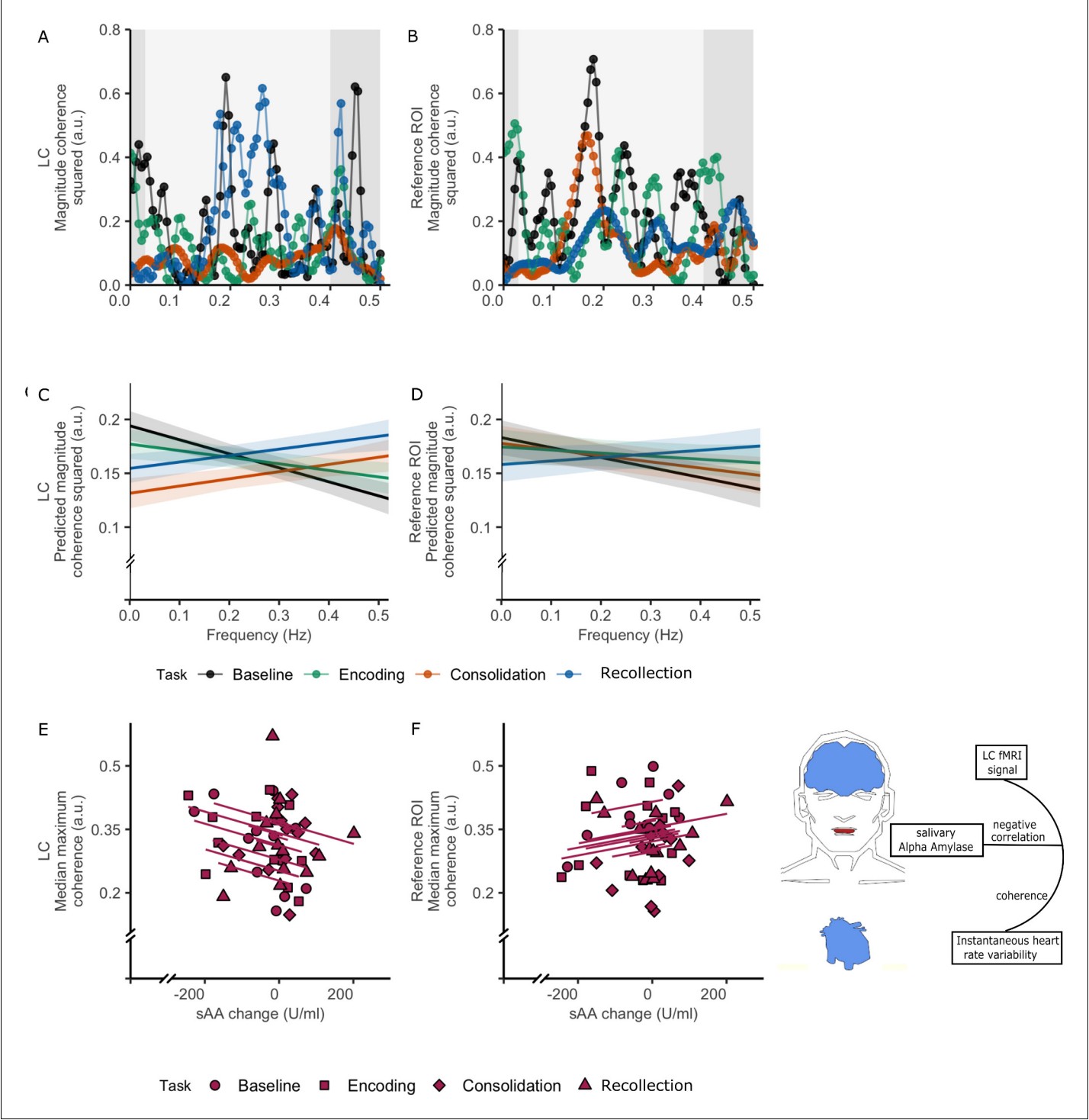

**Figure 5.** Coherence between LC and HRV across the task stages. (**A**) LC coherence magnitude squared plotted against frequency for each task stage for a single participant. The high-frequency HRV band is depicted in a lighter shade. (**B**) Reference ROI-HRV coherence magnitude squared for the same participant. (**C**) The association between frequency and the predicted magnitude squared coherence between the LC and HRV across the various task stages (based on linear mixed effects models, N = 24 (7920 observations), p-values for all comparisons are provided in *Supplementary file 7*). Shaded regions in C and D show the 95% confidence intervals. (**E**) Repeated correlations between LC-HRV coherence and ΔsAA ($R_{rm}$(df)=−0.383 (34), p-value=0.021). (**F**) Reference ROI-HRV coherence and sAA change levels (not significant, $R_{rm}$(df)=0.218 (34), p-value=0.201).

The online version of this article includes the following source data and figure supplement(s) for figure 5:

**Source data 1.** Coherence between locus coeruleus BOLD and HRV.

*Figure 5 continued on next page*

*Figure 5 continued*

**Figure supplement 1.** Coherence between LC and HRV across the task stages for the spatially normalized pipeline.

**Figure supplement 2.** Coherence between LC and HRV across the task stages for the FIXed pipeline Note: Coherence between LC and HRV across the task stages.

**Figure supplement 3.** Coherence between LC and HRV across the task stages for the FIXed + Explicit Resp pipeline.

trials during high arousal (High Arousal successful Encoding > High Arousal forgotten, peak z-value = 2.9, *Figure 6D*) and bilaterally for encoding of emotional stimuli relative to neutral during high arousal (High Arousal Emotional Encoding > High Arousal Neutral Encoding, peak z-value = 5.8, *Figure 6E*).

The extracted beta-coefficients of this contrast correlated with emotional memory performance (r = 0.506, N = 24, p=0.019), indicating that higher LC activity in emotionally encoded trials under high arousal as compared to neutral trials was related to better memory performance. These results suggest that the MTL and BLA is involved in the accuracy of learning, while LC clusters are involved in encoding of emotional stimuli, a role that is particularly modulated by arousal.

## The hippocampus predominates successful recollection

We expected substantial involvement of the hippocampus during successful recollection, but potentially also involvement of the LC and amygdala during emotional recollection. The GLM revealed activation in the left CA1 and subiculum for successful recollection (*Supplementary file 8B*, *Figure 6F*). We found no significant clusters for the emotional versus neutral contrast or for the interaction between recollection success and valence (emotional successful recollection >neutral successful recollection). While we did see a cluster encompassing voxels in the LC region for the emotional vs. neutral and emotional successful recollection >neutral successful recollection contrasts, this cluster contained voxels in the 4th ventricle and the peaks of the clusters were not located in the LC and therefore should be interpreted with caution. No clusters were observed in the MTL or pons related to the HRV local minima or when convolving arousal with the memory regressors. Therefore, involvement of the LC during recollection could not be confirmed in our data.

## Interactions between the LC and MTL transition across task stages

To compare interactions between regions across the task stages we performed seed-to-voxel analyses using cross-correlation at lag = 0 s (denoted as functional connectivity FC) between the LC using the component of the baseline resting-state (*Figure 4—figure supplement 1*) and the predefined MTL regions (*Figure 6—figure supplement 1*): hippocampus (HIPP), amygdala (AMY) and entorhinal cortex (EC). We observed a decrease in FC during consolidation compared to baseline between the LC and left BLA, (para)subiculum and EC (*Supplementary file 8C*, *Figure 7A*). No effects were observed when comparing baseline with encoding or recollection and these findings were reproduced across preprocessing pipelines (*Figure 7—figure supplement 1*).

Areas related to arousal and salience processing, such as the LC, have particularly variable behavior that can result in a negative FC value, complicating their interpretation (*Chang and Glover, 2010*). As our FC analyses assume stationarity within the fMRI timeseries, we aimed to understand these negative FC changes by exploring the frequency-specific magnitude and phase correlation changes in the follow-up spectral coherence analyses. Phase or magnitude changes in the frequency plane can result in reduced cross-correlation values.

To understand whether timeseries from specific regions were leading or lagging during the task stages, we estimated the median phase lag and coherence across the frequency band of maximum coherence between the timeseries of the LC and MTL regions. Similar to previously determined LC-HRV median maximum coherence (*Figure 5*), we calculated the confidence interval of the median of the maximum-individual coherence between timeseries. Linear mixed effects models were implemented with either median coherence or phase at maximum coherence between the task changes as outcome and task stage, timeseries combination and their interaction as fixed effects. We observed an increase in HIP-AMY coherence for encoding compared to baseline (β = −0.13, t(821)=-4.39, p<0.001, 95% CI[−0.20,−0.052]), a decrease in recollection compared to encoding (β = 0.10, t(821)=3.37, p=0.004, 95% CI[0.023, 0.171]) and a trend for consolidation decrease compared to

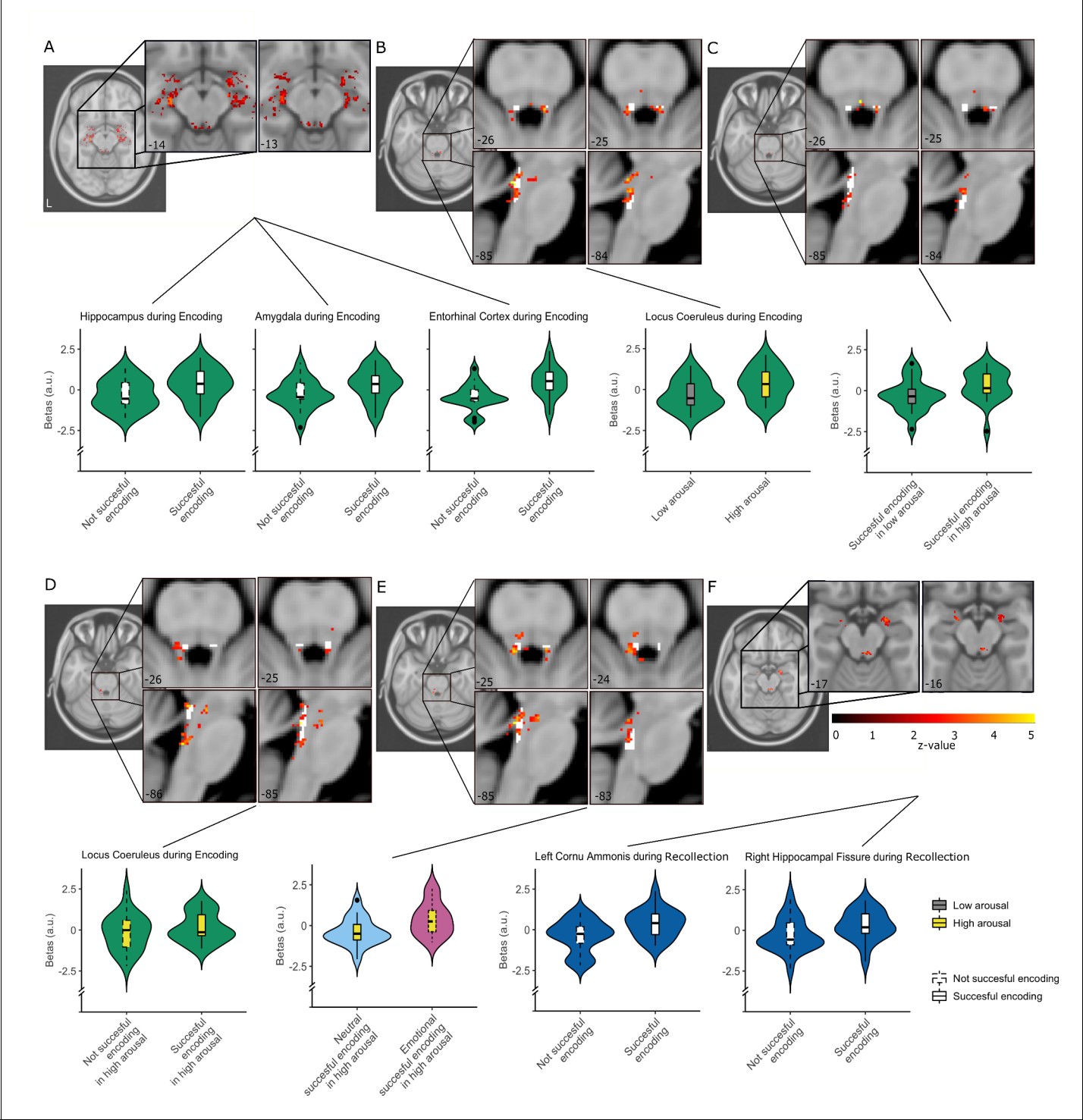

**Figure 6.** LC and MTL activity during the task stages (N = 24). (**A**) Bilateral hippocampus, bilateral amygdala and entorhinal cortex activation during encoding (Encoding >Not Encoding; thresholded at p<0.05, uncorrected). (**B–E**) LC activation during encoding (z-values thresholded at p<0.05, uncorrected). (**B**) LC activation during successfully encoded events with high arousal (HRV minima) (High Arousal successful encoding >Low Arousal successful encoding). (**C**) LC activation during High Arousal Encoding >Low Arousal Encoding and during D) High Arousal successful Encoding >High Arousal forgotten. (**E**) Activation of LC for emotionally high arousal encoding events (High Arousal Emotional Encoding >High Arousal Neutral Encoding). (**F**) Bilateral CA1 activation during recollection of correctly encoded trials (Successful recollection >Forgotten recollection) (z-values thresholded at p<0.05, uncorrected). Boxplots inside the violinplots show median, quartiles (boxes) and range (whiskers). The layer containing white

*Figure 6 continued on next page*

*Figure 6 continued*

voxels demonstrates the location of the Keren template as reference (**Keren et al., 2009**). Individual brainstem results for the encoding contrasts are shown in **Figure 6—figure supplement 2**.

The online version of this article includes the following source data and figure supplement(s) for figure 6:

**Source data 1.** Locus coeruleus and medial temporal lobe activity during the task stages.
**Figure supplement 1.** Mask of regions used in the GLM and functional connectivity analyses.
**Figure supplement 2.** Individual brainstem results for the GLM contrasts during encoding.
**Figure supplement 3.** Encoding-related brain activation for recognition contrasts.

baseline (β = −0.07, t(821)=-2.35, p=0.088 95% CI[−0.141, 0.007]) and a change in phase from AMY leading the LC in encoding to the LC leading the AMY in recollection (β = 0.91, t(821)=2.73, p=0.033, 95% CI[0.051,1.772] *Figure 7B*). Coherent behavior between timeseries may be influenced by other brain regions. Therefore, we calculated partial coherences and compared these against a null distribution consisting of partial coherences from randomized time-series with matched auto-regressive coefficients. The surviving significant coherent relationships between timeseries are visualized in *Figure 7 (B–E)* in terms of coherence magnitude and phase. Accounting for timeseries of the AMY reduced HIPP-LC coherence during encoding, and accounting for HIPP timeseries significantly reduced coherence between AMY and LC during consolidation.

When relating emotional memory to coherence for these ROI pairs using linear regressions (adjusted for baseline coherence), we observed that greater memory performance was associated with greater AMY-EC coherence during encoding (β = 0.28, t(21)=2.62, p=0.038, 95% CI [0.057,0.508]), and EC-HIP coherence during encoding (at trend level: β = 0.27, t(21)=2.38, p=0.054, 95% CI[0.033,0.512]).

## LC structure is associated with sAA and memory performance

Previous studies related LC structure (intensity) to cognitive and autonomic tone measures (**Hämmerer et al., 2018**; **Mather et al., 2017**). The LC segmentation is depicted in *Figure 8A–B* (see also *Figure 8—figure supplement 1*). The LC TFL intensity normalized to a pontine tegmentum ROI showed a positive correlation with emotional recollection scores (r = 0.56, p=0.004, *Figure 8D*). Increased motion correlated borderline with TFL intensity ratio (r = 0.38, p=0.071) and emotional recollection (r = 0.34, p=0.108), but adjusting for motion did not alter the positive relationship between normalized TFL intensity and emotional recollection (p=0.004). When subdividing the LC into three equisized parts (*Figure 8C*), we observed a positive correlation with emotional recollection (*Figure 8D*) for the medial (r = 0.52, p=0.009) and caudal (r = 0.43, p=0.034) parts of the LC, not rostral (r = 0.37, p=0.074).

We then related the mean TFL intensity ratio to our measure of autonomic tone and observed a positive relation with ΔsAA during consolidation (r = 602, p=0.008, N = 18). When examining the different sections of the LC, this positive relationship was only observed for the caudal part of the LC (r = 0.79, p<0.001, N = 18). No association was observed between TFL intensity and rMSSD or LC BOLD variability.

## Discussion

The LC is connected with evolutionary ancient brainstem regions involved in autonomic functions, but also with the associative neocortical regions, allowing it to be 'the cognitive limb of a globally conceived sympathetic nervous system' (**Aston-Jones et al., 1991**). We set out to investigate the contribution of the LC and its interactions with MTL regions during arousal-related memory enhancement by combining dedicated ultra-high-field MRI methods, an established emotional memory task, physiological and saliva measures. Consistent with previous work, we observed that forming adaptive memories in response to arousing events was predominantly associated with NE- activity. Considerably less is known about the LC's interactions with MTL regions during the different memory processing stages. Our results now demonstrate that LC's role during these memory processing stages was shaped by arousal-related autonomic changes and its close interactions with the amygdala. During encoding, stimuli eliciting arousal-related autonomic changes were associated with LC activity following amygdala activation during successful learning. During consolidation, the role of

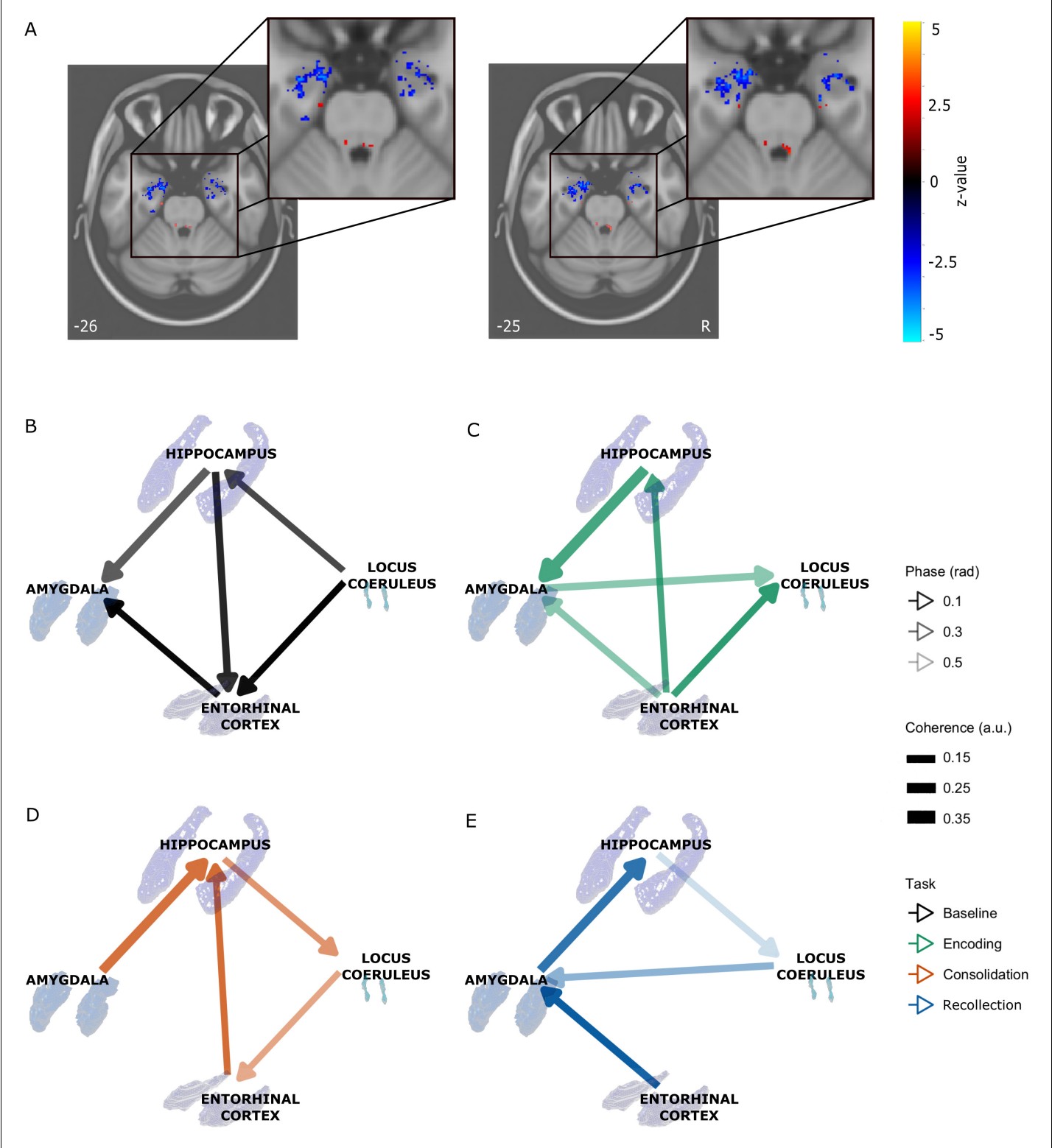

**Figure 7.** Integration of LC and MTL timeseries across the task (N = 24). (**A**) Functional connectivity (FC) from LC component to MTL structures across task stages. Clusters of decreased FC in hippocampus, amygdala and entorhinal cortex were observed comparing consolidation to baseline (p<0.05, uncorrected). (**B-E**) Median maximum coherence and phase across task stages between timeseries from the LC and the clusters from the hippocampus, amygdala and entorhinal cortex from the FC analysis (**B**: baseline resting-state; **C**: encoding; **D**: consolidation; **E**: recollection). Confidence intervals of maximum coherence were calculated per timeseries and the median maximum coherence and phase were extracted. The significant partial coherence

*Figure 7 continued on next page*

*Figure 7 continued*

(regressing out timeseries from other regions) is visualized. The thickness of the edges indicates the strength of coherence. The absolute phase difference between timeseries is shown in transparencies. Arrows indicate the direction of the phase lag from leading to lagging.

The online version of this article includes the following source data and figure supplement(s) for figure 7:

**Source data 1.** Coherence and phase between time series of the locus coeruleus and medial temporal lobe clusters.

**Figure supplement 1.** Functional connectivity from LC to MTL during consolidation compared to baseline across preprocessing steps.

the LC gradually changed from being arousal-driven to maintenance or updating internal models during recollection, involving predominantly hippocampal activity. These results highlight the dynamic nature of neuromodulatory brain networks during arousal-related memory processes (*Hermans et al., 2011*). Understanding the neural mechanisms underlying memory for emotional experiences increases our understanding of diseases in which a dysregulation or decoupling of one of these systems could lead to affective, autonomic or neurocognitive disorders (*Samuels and Szabadi, 2008b*; *Jacobs et al., 2019*).

Activation of the LC occurs in parallel with the autonomic system, and modulates cognition: across the entire memory task greater variability in LC BOLD signal was associated with lower parasympathetic activity (rMSSD), which in turn was associated with greater sympathetic modulation (ΔsAA) and better memory performance. Beyond function, we also observed that structural properties of the LC, the middle section, related to better emotional memory performance, and greater caudal LC intensity was related to greater NE. These correlations fit with the well-described topographic projections of the LC, where the rostral and middle part project densely to higher-order associative regions of the brain and the caudal part of the LC to the nucleus ambiguus and dorsal motor nucleus of the vagus (*Sara, 2009*; *Samuels and Szabadi, 2008a*), which both modulate the parasympathetic outflow based on sympathetic signals.

We also observed task-stage-specific associations between LC's connectivity to the MTL regions and autonomic tone measures. During consolidation, we observed a different LC-MTL functional connectivity coupling relative to baseline, possibly reflecting transient processes related to less parasympathetic tone (lower coherence in the high frequency domain). Correspondingly, increases in sAA during consolidation correlated positively with emotional performance. Such increases in sympathetic tone may reflect arousal-related NE-synaptic consolidation processes (*Packard et al., 1994*).

While systems consolidation requires years, synaptic consolidation represents the initial phase after learning where mnemonic representations change and new memory tracers are formed. Previous reports demonstrated that synaptic consolidation may be captured by the resting-state signal after learning a task (*Vilberg and Davachi, 2013*; *Tambini et al., 2010*; *Jacobs et al., 2014*). This consolidation phase may be a transitional stage, where the role of the LC slowly alters from an arousal-based learning experience to maintenance of memories via long-term potentiation (LTP). LTP depends on the activation of the BLA, highlighting the critical role of the amygdala (*Frey et al., 2001*). A myriad of animal studies have demonstrated that elevated NE in the amygdala affects memory consolidation for stress-induced events and can facilitate hippocampal synaptic plasticity (*Hermans et al., 2014*; *McIntyre et al., 2003*).

In support of this premise, we observed mainly arousal-related LC activity during encoding. During encoding, the LC was activated during (successful) acquisition of information that was associated with elevated arousal. However, amygdala and hippocampal activation was only detected for successful learning, independent of valence of arousal. Arousal-related amygdala activation may have been attenuated by the cognitive task during encoding (*Lange et al., 2003*). Nonetheless, arousal-related LC-activation with neural input from amygdala-entorhinal co-activation was associated with memory enhancement. This is largely consistent with work reporting that sympathetic arousal can predict memory performance for high-arousing stimuli when this coincides with amygdala activation (*de Voogd et al., 2016*). During recollection, we observed no arousal-related activity, but successful recollection was associated with CA1 activity. Activation of β-adrenoreceptors in the CA1 during consolidation is required for later retention of the memory task, but not for the aversive component of the task (*Mello-Carpes and Izquierdo, 2013*; *Chen et al., 1992*). The markedly higher frequency window of the HRV-LC coherences during recollection, suggests less NE-sympathetic

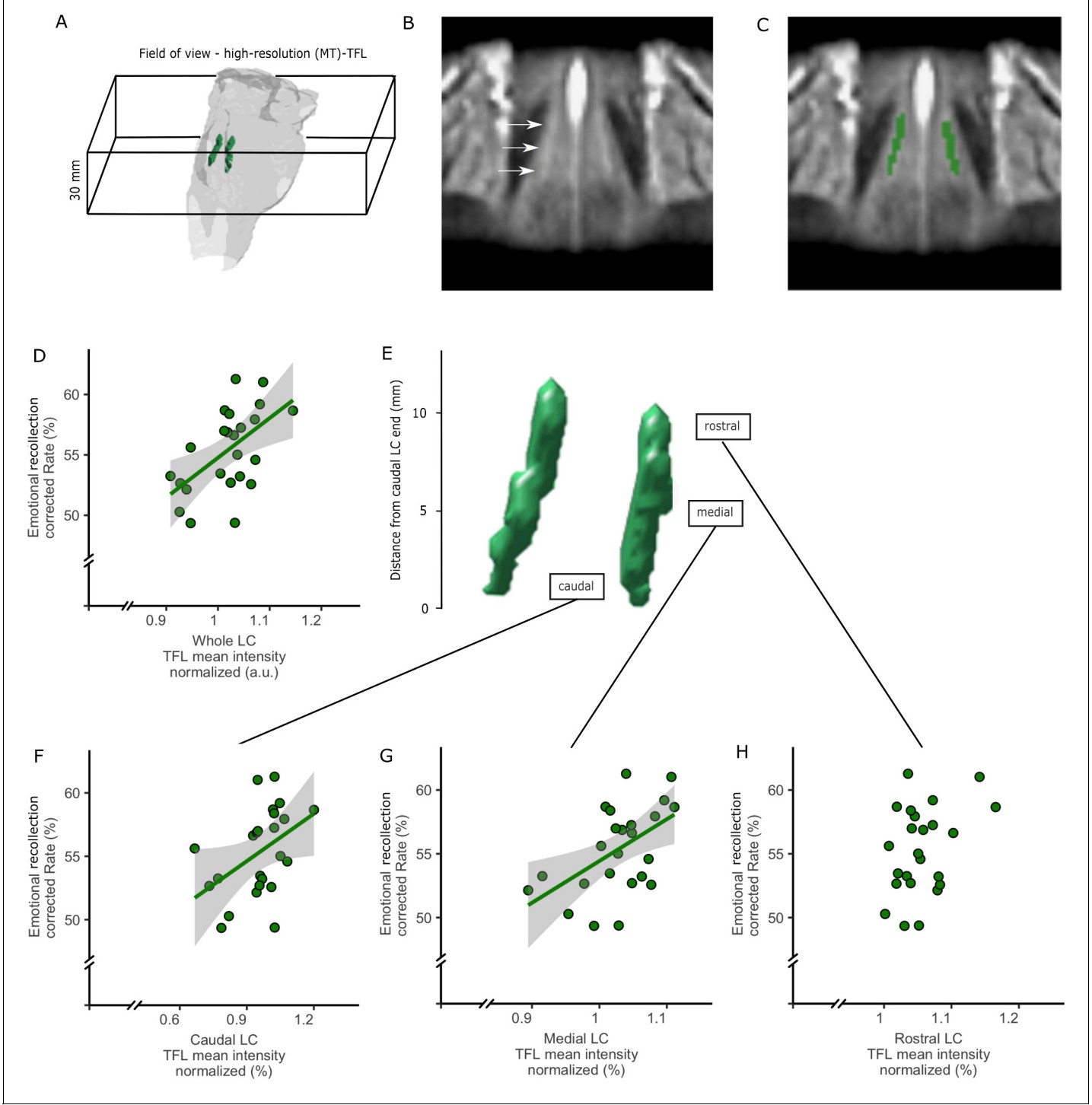

**Figure 8.** Structural segmentation of the LC and associations with emotional memory performance (N = 24). (**A**) 3D visualization of the LC placement in the pons. (**B**) The LC template, coronal view (hyperintense rod-like shape). (**C**) Segmentation of the LC in template (green). (**D**) TFL mean intensity normalized with a pontine reference region of the entire LC correlated with emotional memory recollection rate (r = 0.562; p-value=0.004) E) The LC was divided into three equisized subsegments. (**F–H**) LC segments intensity correlation with emotional memory recollection rate: (**F**) caudal (r = 0.523; p-value=0.034) (**G**) medial (r = 0.523; p-value=0.009) and (**H**) LC rostral (r = 0.523; p-value=0.074). Shaded regions depict the 95% confidence interval. The online version of this article includes the following source data and figure supplement(s) for figure 8:

**Source data 1.** Structural locus coeruleus intensity and memory performance.
**Figure supplement 1.** Template of the locus coeruleus for structural analyses.

responsiveness at this stage. Increases in LC-hippocampus and LC-amygdala coherence variability during recollection are possibly related to increased parasympathetic inhibitory modulation supporting cognitive operations. Alternatively, it could be speculated that spill-over from NE- sympathetic processes during consolidation drive retention processes.

In addition to different activation patterns across stages, we also observed a switch in the role of LC's connectivity with MTL regions. During encoding, the LC is on the receiving end, in particular from the amygdala, whereas this changes to a leading role in recollection. This phase switch between the LC and amygdala conforms with functional and anatomical descriptions of projections between the LC and central amygdala or BLA. The LC receives afferents from the central amygdala mainly at the periLC-region (*Bergado et al., 2007*). The central amygdala controls stress responses and is involved in salience processing and attention processes, important for efficient encoding (*Phelps and LeDoux, 2005*; *Roozendaal et al., 1996*). While the central amygdala has no effect on hippocampal LTP, the BLA has memory-enhancing effects on the subiculum and CA1 via β-adrenergic mechanisms (*Phelps and LeDoux, 2005*; *Roozendaal et al., 1996*). The NE-input to the BLA arrives mainly from the LC (*Chen and Sara, 2007*). The BLA does not project directly to the LC, but via a small direct projection to the central amygdala (*Bouret et al., 2003*).

Recent general theories of brain function, such as the active inference theory (*Friston et al., 2015*), posit that NE responses from the LC can be considered as a prediction error during learning originating from information from the amygdala, whereas signals from the LC to the amygdala and cortex guide transfer of information and model updating (*Sales et al., 2019*). In these models, the amygdala is critical to maintain allostasis. The central amygdala network would control the arousing physiological responses through its descending predicting signals to the brainstem, whereas the BLA would be part of a neutral signal uncertainty network as it receives prediction errors from the LC. It has been suggested that NE increases accuracy of episodic memories (*Barsegyan et al., 2014*). While our study was not designed to test the active inference theory, the proposed models are consistent with our phase-differences across stages. Future in-vivo recordings, computational and carefully designed studies are warranted to test these models in detail.

No study is without limitations. First, as our focus was on the LC and MTL, we specifically opted to acquire fMR images with a high isotropic spatial resolution. This limited our field of view and therefore, we did not investigate distant modulatory effects. Second, the LC is challenging to investigate in humans due to limits in spatiotemporal resolution and lower signal-to-noise ratio in the brainstem compared to neocortical signal. We had to eschew cluster-based correction. The small in-plane diameter of the LC means that only low-levels of smoothing can be applied, which would reduce the signal-to-noise, and the probability of discovering a contiguous cluster in the highly-elongated axial plane of the structure. Ultimately, this would result in an overly stringent multiple comparison adjustment. Nevertheless, we are confident that our GLM findings do not represent false positives due to their reproducibility across preprocessing pipelines, correlation to behavior and consistent anatomical co-localization with an ex-vivo validated template and anatomical atlases.

To conclude, interactions between the NE-LC, amygdala and hippocampus, drive arousal-related memory processing, but transitions in interactions occur along the memory processing stages. During successful encoding, we observed high-arousal-related LC activation, possibly reflecting attentional processes, in conjunction with amygdala activation. During consolidation, LC-MTL connectivity is dynamic and associated with NE increases possibly reflecting LTP. This gradually transitions to learning and updating internal models during recollection, which mainly involves hippocampal activity. These findings contribute to our understanding of LC activation for forming adaptive memories of emotional experiences and can serve as a basis for understanding how LC dysregulation may potentially disrupt this adaptive behavior.

## Materials and methods

### Participants

In total, 27 healthy young right-handed adults (13 female, age 20–30 years old, mean ± SD = 22.95±1.96) were recruited via local advertisements. All participants were screened to exclude a history of major psychiatric or neurological disorders, having a history of brain injury of brain surgery, taking medications that may influence cognitive functioning or being not eligible for

MRI-scanning. All participants had normal or corrected-to-normal visual acuity and reported normal hearing abilities. Since depression has been linked to LC atrophy, all participants were screened on the Hamilton Rating Scale for Depression (all within normal range = 0–10; mean ± SD = 3.1±3.3). All participants also endorsed a degree of appraisal of stress-full events that was within the normative range on the Perceived Stress Scale-10. The Perceived Stress Scale-10 is a questionnaire tapping into global non-specific appraisal of stress that also relates to glucocorticoid levels and the number of stressful life events. All participants received monetary compensation for their participation and provided written informed consent. Approval of the experimental protocol was obtained from the local ethical committee of the Faculty of Psychology and Neuroscience at Maastricht University.

## Neuropsychological assessment

All participants underwent a comprehensive battery of cognitive tests covering episodic memory (15-word learning test (WLT): learning and delayed recall), working memory (digit span forward and backward), language (semantic and category verbal fluency), attention (concept shifting task (CST)), information processing speed (Letter Digit Substitution Test (LDST)) and executive functions (Stroop Color Word Task (SCWT)). All cognitive tests were presented in the same order to each participant.

## Experimental paradigm

To investigate emotional memory a well-established face-name association task (*Sperling et al., 2001*) was modified by including faces with negative or neutral valence.

## Stimuli selection

Several databases consisting of emotional and neutral faces (Ekman, PERT96, ER40, MMI facial expressions, Radboud faces, Utrecht ECVP) (*Ekman, 1992*; *Gur et al., 2002*; *Langner et al., 2010*) were concatenated. All faces were transformed to gray-scale and resampled at the same resolution ($750 \times 830$ pixels$^2$) using Matlab and were manually cropped to the same template and resolution. Distinctive details beyond the face were darkened manually. Pictures that still contained potential distinctive features, such as facial hair, were excluded. All images had the same mean luminance and were presented on a black background. An independent sample of 10 individuals between 20–30 years old (five women) were shown the faces and asked to rate the emotional valence of each face on a 5-point Likert Scale (0: completely neutral, 5: very emotional). These ratings were ranked and 135 faces were selected (68 faces (50% male) with the highest emotional rating, 68 faces (50% male) with the lowest emotional rating (neutral)). Two Dutch native speakers (NV and HILJ) matched these 135 faces to an age-appropriate name from a list of the 400 most popular Dutch names (http://www.behindthename.com/top/lists/netherlands/).

## Task description

The face-name association task (event-related design) consisted of encoding, consolidation, recognition and recollection stages. During encoding the participants were shown 90 faces (45 emotional, 45 neutral, gender-matched across conditions) along with their name for 4 s. The participants were asked to memorize the face-name association. To keep their attention to the task they were also asked to make a subjective judgment as to whether the face matched the name or not. A jitter of average length of 2 s (0.5–6 s, created with optseq [*Dale, 1999*]) was entered between trials, during which a fixation cross was shown. During the consolidation phase, which lasted approximately 8 min, the participants were asked to stare at a fixation cross. During the recollection stage, the participants were shown the 90 faces shown in encoding along with 45 new faces, matched for sex and emotion intensity to the initial 90 ones in a random order. The participants had 3 s to decide whether or not they had seen the face during encoding (recognition phase). If participants indicated they had seen the face before, they subsequently were asked to choose the name that was previously associated with that face (participants received three options, recollection phase). The three names shown were chosen randomly from the list of sex-specific names that were already shown in encoding, to avoid easy name recognition. Recognition or familiarity represents a fast process, the feeling that the face was seen previously without recollecting it. In contrast, recollection is a slow process that involves the active retrieval of details of the face-name pair (*Yonelinas et al., 2010*).

A jitter of average length of 1.5 s (0.5–6 s) was employed between trials, during which a fixation cross was shown. No more than two consecutive faces of emotional valence were shown during any of the task stages. Stimuli were viewed on a back-projection display via a mirror mounted onto the head coil. Behavioral responses were collected through a MR-compatible button box (Current Designs, eight-button response device, HHSC-2 × 4 C). All stimuli, event identities and timings were presented and logged using E-Prime 2.0 (Psychology Software Tools).

We calculated hits and false alarms for both the emotional and neutral condition during encoding. For the recollection (identification of the name), we calculated hits and false alarms for both the neutral and emotional condition. Reaction times were also analyzed. Performance in the recollection condition may be affected by the response bias in the recognition phase between emotional and neutral stimuli. Therefore, we calculated the likelihood ratio beta, as a measure of the response bias (beta = exp (d' x c)) (d' index calculated as by Stanislaw and Todorov, but corrected for ceiling effects [*Jacobs et al., 2015*]). This response bias was derived from the signal detection theory, a method that allows us to differentiate the signal from the noise (*Stanislaw and Todorov, 1999*). The bias measure indicates the extent to which a certain type of response is more probable than another, such as for example responding in a more conservative manner. We performed a paired t-test to compare the bias between emotional and neutral during the recognition phase. This bias was then regressed our from the emotional and neutral recollected hits in two separate regressions (*Figure 2*). The residuals of each regression represented the bias-corrected emotional and neutral recollection rate. Finally, to obtain an individual emotional recollection rate, we calculated an adjusted emotional memory performance score $= \frac{Emotional\ retrieval\ rate}{Emotional\ retrieval\ rate + Neutral\ retrieval\ rate}$ (*Figure 2—figure supplement 1*). This measure was used throughout the manuscript.

## Imaging parameters

MR scans were performed in a 7T Magnetom Siemens (Siemens Healthineers, Erlangen, Germany) with a 32-channel head coil (Nova Medical, Wilmington, MA, USA). First, we acquired a Magnetization Prepared 2 Rapid Acquisition Gradient Echoes (MP2RAGE) sequence (*Marques et al., 2010*) for whole brain imaging (TR = 5000 s, TE = 2.47 ms, flip angle = 5°/3°, voxel size = 0.7×0.7 × 0.7 mm$^3$, number of slices = 240). An in-house developed magnetization transfer-weighted turbo flash (MT-TFL) sequence sensitive to LC contrast (*Priovoulos et al., 2018*) was performed to image the LC at high resolution. The sequence consisted of a multi-shot 3D readout (TR = 538 ms, TE = 4.08, flip angle = 8°, voxel size = 0.4×0.4 × 0.5 mm$^3$, number of slices = 60) with center-out k-space sampling, preceded by 20 long off-resonant Gaussian sinc pulses (pulse length = 5.12 ms, bandwidth = 250 Hz, B$_1$ = 0.25µT). For the MT-TFL sequence, the field-of-view (FOV) was placed approximately perpendicular to the pons and covered the area between the inferior colliculus and the inferior border of the pons. A matched TFL sequence but with the MT pulses turned off was similarly acquired. For high resolution BOLD fMRI imaging we acquired multiband EPI sequences (*Moeller et al., 2010*; *Setsompop et al., 2012*) (TR = 2000 ms, TE = 19 ms, isotropic voxel size = 1.25×1.25 x.125 mm$^3$, number of slices = 50, M-factor = 2, GRAPPA R = 3). The field of view was placed at an angle of 45° to the brainstem, to reduce the effect of physiological movement at the pons level and to be approximately perpendicular to the hippocampus to optimize subfield differentiation. After each BOLD fMRI acquisition, five more volumes were acquired with reversed phase encoding direction to facilitate distortion correction. Four functional runs were collected: initial resting-state, an encoding phase, second resting-state and a recognition phase.

## Saliva samples collection

Salivary alpha-amylase (sAA) is a sensitive marker for sympathetic nervous system reactivity that correlates with plasma norepinephrine and shows similar dynamics (*Rohleder and Nater, 2009*). Samples of sAA were collected at various time points throughout the scan session: before entering the bore, before the first resting-state scan, before encoding, after encoding, before recollection, after recollection and 30 min after the scan session. This sampling scheme allows us to measure acute responses. Saliva samples were obtained with cotton swabs (Sarstedt, Nümbrecht, Germany) that were exchanged with the participants via a MRI-compatible arm extension. Each test session was scheduled at the same part of the day. Participants were instructed to refrain from food consumption or brushing their teeth one hour before scanning and to abstain from caffeinated drinks for four

hours prior to the scanning. Compliance was checked with a questionnaire and all participants followed the instructions. Before the first saliva sample the participants were asked to rinse their mouth with water. The participants were instructed to place the cotton roll between their cheek and their teeth or beneath their tongue and to not chew on it; after one minute the cotton rolls were retrieved. The saliva samples were stored at −20°C until their processing at the Dresden University of Technology. After thawing, saliva samples were centrifuged at 3000 rpm for 5 min. Alpha-amylase was measured by a quantitative enzyme kinetic method (*Nater and Rohleder, 2009*). When inadequate amount of saliva was collected at specific time points, these time points were excluded from the saliva analysis. Missing data for the sAA time point are provided in *Supplementary file 2*.

## Pulse rate preprocessing

Heart rate and chest movements were measured concurrently during the fMRI acquisitions with a pulse oximeter and respiratory bellows respectively. The pulse rate signal was preprocessed in R with the package RHRV. Automatic removal of outliers was performed by adaptive thresholding (beats whose value exceeded the cumulative mean threshold were rejected) and by rejecting beats with physiologically improbable values (less than 25 and more than 200 beats-per-minute) (*Vila et al., 1997*). To allow power spectral analysis, we interpolated to an evenly spaced pulse rate series with a cubic space interpolation and pulse (heart) rate variability (HRV) timeseries (*Hill and Siebenbrock, 2009*; *Pinheiro et al., 2016*) (differences of successive R-R intervals, sampled equidistantly at the TR) were extracted. Root-mean square differences of successive R-R intervals (rMSSD) were calculated as a measure of high-frequency (0.15–0.40 Hz) arousal over each time period. rMSSD has been shown to tightly correlate with parasympathetic activity and be relatively free of respiratory influences (*Hill and Siebenbrock, 2009*). Lower rMMSD is associated with poor vagus-mediated HRV and indicative of higher stress of arousal. For the analyses focused on the time-domain, we used the rMSSD metric and for the frequency domain, we used the HRV terminology. Datasets with missing cardiac physiological signal for more than 10% of the time points were excluded (*Supplementary file 2*).

## Preprocessing of MRI data

### Preprocessing of structural data

The T1-weighted MP2RAGE images were processed using FreeSurfer (FS) version 6.0.0 (https://surfer.nmr.mgh.harvard.edu/) using the software package's default, automated reconstruction protocol as described previously (*Fischl et al., 1999*). Briefly, each T1-weighted image was subjected to an automated segmentation process involving intensity normalization, skull stripping, segregating left and right hemispheres, removing brainstem and cerebellum, correcting topology defects, defining the borders between grey/white matter and grey/cerebrospinal fluid and parcelling cortical and subcortical areas. The hippocampal-amygdala segmentation algorithm in FS 6.0 predicts the location of subregions by using a probabilistic atlas built from a combination of manual delineations of the hippocampal formation from ultra-high resolution ex-vivo MRI scans showing definitive borders and manual annotations of the surrounding subcortical structures (e.g. amygdala, cortex) from an independent dataset (*Iglesias et al., 2015a*). Using FS's visualization toolbox, freeview, we visually inspected and, if necessary, edited each image for over- or under-estimation of the gray/white matter boundaries and to identify brain areas erroneously excluded during skull stripping. In addition, we checked that the hippocampal/amygdala subregion mask was well positioned and that the ranking of subfield-specific volumes was consistent with the literature. An automated segmentation of the pons of the brainstem with Bayesian inference was applied (*Iglesias et al., 2015b*). The T1 to MNI template registrations were calculated using a diffeomorphic transform with the antsRegistrationSyN.sh function (ANTS 2.1 (http://stnava.github.io/ANTs/) [*Avants et al., 2011*]). This algorithm has been shown to provide superior performance compared to most other registration algorithms used in neuroimaging (*Avants et al., 2011*; *Klein et al., 2009*).

The partial field-of-view high resolution MT-TFL scans were registered to the T1 data using the boundary-based registration. A study-specific template of the LC scans was created with an iterative diffeomorphic warp estimate using the buildtemplateparallel.sh script of the ANTS package. The LC was segmented on the template (by NP). This segmentation was repeated 4 weeks later and only voxels that were identified during both sessions were used in the final segmentation. The left LC was

12 mm in length (108 voxels, 9.918 mm$^3$) while the right LC measured 10 mm in length (103 voxels, 9.459 mm$^3$). The LC was divided in three equisized segments (rostral, medial and caudal) for further analyses. The segmented LC was projected to the high-resolution LC specific sequences, including MTR, MT-TFL and TFL to extract the mean intensity. MTR was calculated as

$$MTR = \frac{MTweighted - TFL}{TFL}$$

The reference region for normalization was defined as a $10 \times 10$ voxels ROI in the pontine tegmentum at the same level as the LC). Normalization was done by dividing the LC intensity values with the reference region intensity. Given that motion affects MR images, a retrospective motion metric (AES) was also calculated with the homonymous Matlab toolbox.

## Preprocessing of BOLD fMRI data

Image preprocessing of the fMRI data was performed with FSL 5.0.9 (https://fsl.fmrib.ox.ac.uk/fsl/fslwiki/). The first five volumes were removed to ensure that data within steady-state were examined. The functional images were motion-corrected to the last image with a 6-dof transform using MCFLIRT (*Jenkinson et al., 2002*). To reduce EPI-distortions a displacement field was calculated and applied with FSL-TOPUP by using the last 5 frames of each fMRI scan and the following reversed phase encoding direction scans. As the brainstem is known to be sensitive to physiological artefacts, we prepared four separate denoising pipelines, to allow us to check the reproducibility of our findings (*Figure 1—figure supplement 2*). The pipelines consisted of 1) minimally processed data (distortion-, motion- and slice-timing corrected; denoted as 'Spatially normalized' data), 2) spatio-temporally denoised data (distortion-, motion-, slice-timing corrected and with the ICA-FIX algorithm applied (*Griffanti et al., 2014*) and smoothed; denoted as 'FIXed' data), 3) spatio-temporally denoised data along with respiration temporal regressors (distortion-, motion-, slice-timing, FIXed corrected and with respiration regressors regressed out with PNM and smoothed; denoted as 'FIXed + explicit Resp' data) 4) spatio-temporally denoised data along with respiration and pulse temporal regressors (distortion-, motion-, slice-timing, FIXed corrected and with both respiration and pulse regressors regressed out and smoothed; denoted as 'FIXed + explicit Phys' data). To apply the FIX algorithm, the already available CMRR-based 7 T HCP dataset with a similar multiband factor and resolution was used as a trained-weight file; the noise component classification was checked against manual classification performed by NP and LP. The interrater difference between FIX and the manual classification was smaller compared to the interrater difference between the two manual raters and therefore, FIX was preferred. FIX removed a similar percentage of noise components across the task stages (median baseline: 48.52% [IQR = 41.76–52.45%]; median encoding: 54.47% [IQR = 35.42–64.04%]; median consolidation: 48.93% [IQR = 33.99–59.68%]; median recollection: 49.67% [IQR = 43.74–64.3%]).

The cardiac and respiratory waveforms recorded during the scan session were entered as slice-specific regressors; three orders of cardiac, fours orders of respiratory and a single set of interaction terms were used as suggested by Harvey and colleagues (*Harvey et al., 2008*). Because HRV is known to vary depending on the respiration phase (respiratory sinus arrhythmia) (*Yasuma and Hayano, 2004*), an additional binary slice-specific regressor was added that represented inhalation and exhalation to model BOLD variance related to one of the two phases. The regressors were then used to model the physiological noise in the fMRI signal; the noise signal was adjusted for interactions with the ICA-FIX denoising regressors, so that artefacts were not re-introduced due to the successive regressions. Finally, slice-timing correction was performed based on the multiband acquisition matrix and a 3D spatial smoothing kernel of 1.5 mm FWHM was applied (size of the kernel estimated in accordance with the minimum LC width). Throughout the main manuscript, the FIXed data were employed (since this ensured the maximum possible sample size, due to no missing pulse and respiration data). Results of the sensitivity analyses (reproducibility) by running the other pipelines are indicated were applicable. A boundary-based registration was calculated for the linear registration between the fMRI and T$_1$data using FSL's epi_reg based on the FreeSurfer segmentations. The T1 to MNI template registrations were done using ANTS as described in the preprocessing of the structural data. The linear transformations were transformed to the ITK-SNAP reference using c3d tools and combined with the warps, so that when transforming the data from native fMRI

to MNI there was a single interpolation. The accuracy of the transforms was individually checked by NP (*Figure 1—figure supplement 1*).

## Initial resting-state ('baseline')

To robustly estimate the LC signal, we performed group-level ICA in the baseline resting-state data, which included projection of the fMRI data to the MNI 1 mm template, masking for the brainstem and temporal concatenation, followed by spatial ICA by maximizing non-Gaussian sources. The output group-level component map was thresholded at $p<0.01$ FWE-corrected and binarized. One component showed strong spatial correlation with our LC atlas (2.47% explained variance). We then applied dual regression to estimate the individual LC timelines; statistical significance was set at $p<0.05$ family wise error (FWE)-corrected for multiple comparisons with TFCE (Threshold-Free Cluster Enhancement). A component that explained a similar amount of signal variance and was at a similar location yet spatially unrelated to the LC was picked as an additional control component and the individual timelines were also extracted. Finally, the dual regression (and the extraction of timelines) was repeated with the same component map for all preprocessing stages (spatially normalized (after motion, distortion and slice-timing correction), with FIX applied, with FIXed + explicit Resp applied and with FIXed + explicit Phys) to ensure that the LC response was consistent across different steps of denoising and therefore not the result of physiological noise or the steps taken to remove it.

## Task-related fMRI analyses

At individual level, a small-volume corrected general linear model was fit at each voxel with FMRI Expert Analysis Tool (FEAT), using generalized least squares with a voxel-wise, temporally and spatially regularized autocorrelation model. A high-pass filter (cutoff at 100 s) and minimal spatial smoothing (kernel = 1.5 mm) were applied before the fit. The area of interest (ROI) was defined in advance and consisted of the LC and surrounding region projected to the native fMRI space (LC, HIPP, AMY and EC being our a-priori ROIs; the 4th ventricle and pons were also included to ensure the spatial specificity of the effect and that it was not due to uncorrected motion due to proximity to a CSF space (total mask size in native space approximately = 18000 voxels). The regressors consisted of the emotional trials that were later successfully recollected, the neutral trials that were later successfully recollected, the emotional trials that were not recollected, the neutral trials that were not recollected and their respective convolutions with the HF-HRV local minima. Contrasts of interests were emotional trials vs neutral trials (Emotional >Neutral), successfully recollected trials vs. not successfully recollected (Successful encoding >Not successful encoding) and the interaction (Emotional successful encoding >neutral successful encoding).

The HRV local minima calculated over a rolling window of 15 s was also added as a regressor to account for the main effect of arousal on LC. Additionally, we examined (emotional) encoding during the local HRV over a sliding window of 15 s (high arousal). These regressors were convolved with the high arousal regressor and we contrasted high arousal events with the rest of the timeline (High Arousal vs. Low Arousal), as well their interactions with encoding (High Arousal Successful encoding >Low Arousal Successful Encoding, High Arousal Successful Encoding >High Arousal Not Successful Encoding). To examine the effect of emotional valence on encoding events under high arousal we calculated (High Arousal Emotional Encoding vs. High Arousal Neutral Encoding). These contrasts were examined both in the encoding and recollection task stages.

Contrasts related to recognition were not our primary interest, however, we fit encoding regressors for emotional trials that were later successfully recognized, the neutral trials that were later successfully recognized, the emotional trials that were not recognized and the neutral trials that were not recognized. Contrasts of interests were emotional trials vs. neutral trials (Emotional >Neutral), successfully recognized trials vs. not successfully recognized (Successful recognition >Not successful recognition) and the interaction (Emotional successful recognition >neutral successful recognition). Results are presented in *Figure 6—figure supplement 3*.

All regressors were convolved with a double gamma haemodynamic response function before entered in the model. The contrasts of the coefficients and their variance estimates were subsequently warped to the MNI 1 mm space and a group-level linear mixed effects regression was fit at each voxel, using generalized least squares with a local estimate of random effects variance and outlier de-weighting. The Z-maps were cluster-wise corrected with a cluster significance threshold of

p<0.05 after being thresholded at Z > 2.3. Cluster peaks are reported with respect to the Freesurfer atlas.

To extract beta estimates from the LC, we first dilated the LC-segmentation by 2.3 mm and excluding voxels that overlapped with the 4th ventricle mask. The dilation was performed to match the physiological point spread function (FWHM) of the BOLD effect at 7T for gradient-echo acquisitions (*Shmuel et al., 2007*). Then, the z-map was thresholded at p<0.05 and extracted the mean subject-specific beta values using the dilated LC mask.

## LC Functional connectivity across all memory stages

Seed-to-voxel functional connectivity analyses were performed between the LC and regions of interest in the MTL (hippocampus, amygdala and entorhinal cortices) across all task stages. The LC ROI was defined based on ICA component in the baseline resting-state scan (see 'initial resting-state'). Correlation maps were generated by correlating the average BOLD time course from the LC-seed with the BOLD timeseries of each voxel within our a-priori defined ROIs (hippocampus (HIPP), amygdala (AMY) and entorhinal cortices (EC)). These ROIs were dilated by 2 mm to include surrounding voxels in the ventricles to scrutinize for possible partial-volume or motion-related false-positives in the CSF. The correlation coefficients were converted to normally distributed scores using Fisher's r-to-z transformation. Individual maps were warped to the 1 mm MNI template and fed into the second-level analyses with a voxelwise t-test to compare functional connectivity between the LC and the ROI's comparing encoding, consolidation, recollection with baseline. The second-level statistical analysis consisted of a non-parametric, permutation test with threshold free cluster enhancement; results were family-wise error corrected (p-value<0.05).

## Coherence analysis

Given that autonomic influences in the HRV are known to be most prominent in specific frequency bands (*Shaffer and Ginsberg, 2017*), it is of interest to examine the correlation between fMRI and HRV measures per frequency. The magnitude squared coherence between timeseries can then be expressed as:

$$C_{A,B}(f) = \frac{|P_{A,B}(f)|^2}{P_{A,A}(f)^2 \cdot P_{B,B}(f)^2},$$

with $P_{A,B}(f)|^2$ being the cross power spectral density and $P_{x,x}(f)^2$ being the power spectral density for timeseries $A$ and $B$. We estimated the coherence magnitude squared over 90 frequency bins with Hamming windows (20 % overlap) within the 0 - 0.5 Hz range for the LC and HRV, as well as LC and FC clusters. Similarly, the phase lag between the timeseries per frequency bin was estimated as:

$$\theta_{fMRI,HRV} = tan^{-1}\left(\frac{Re\left(P_{fMRI,HRV}(f)\right)}{Im\left(P_{fMRI,HRV}(f)\right)}\right).$$

To obtain a robust single-point coherence metric, we estimated the median across the frequency band of maximum coherence. This was defined as the confidence interval (CI) around the median of the distribution of individual maximum-coherence frequencies between LC (or reference) and HRV at baseline (given that the frequency of maximum coherence between LC and HRV varies between individuals). These CIs were obtained for all timeseries and we extracted the median coherence and phase lags within the CIs. A coherence-magnitude null distribution was created by fitting an AR1 model for each timeseries, creating a randomized timeseries with a matched AR coefficient and calculating the respective coherences. These null-distributions are similar to the bootstrap and hence, the resampled timeseries have similar properties as the original data (distribution, autocorrelation) and have less bias in the rejection rates of the null-hypotheses.

## Statistical analyses

Statistical analyses were performed using statistical software (R version 3.4.2). Group characteristics are presented in median and interquartile range. Differences between emotional and neutral trials in the behavioral data were tested with Wilcox signed-rank test or robust linear regression using the Huber-M estimator. Robust regression is a more conservative test compared to linear least-square

regression methods, as the resulting models are stout against outliers. Changes in the autonomic measures (sAA and HRV) across the task stages (baseline, encoding, consolidation, recollection) were assessed with mixed effects linear regression using the maximum likelihood estimation with the autonomic measure as outcome measure, stage as predictor and a random intercept for each subject. The linear mixed effects model can account for missing data in longitudinal datasets. The most complex linear mixed effects models in this manuscript follows this formula:

$$Outcome_{ij} = \beta_1 + \beta_2\,PredictorA_i + \beta_3\,Task\,stage_{ij} + [\beta_4\,predictorA_i \times Task\,stage_{ij}] + b_{1i} + \epsilon_i,$$

Outcome = outcome variable measured over time
Predictor A: variable of interest depending on the investigated model
Task stage$_{ij}$ = Task stage (time)
$b_{1i}$ = random intercept for each subject

To obtain sAA measures over the same time frame as the HRV (rMSSD) measure, we calculated ΔsAA as: sAA end - sAA beginning for each task stage. To understand relationships between ΔsAA, HRV and LC activity across the task stages, we performed bootstrapped (n = 5000) repeated measures correlation.

For the frequency-specific effects of task stage on LC-HRV coherence, we entered the coherence magnitude squared into a linear mixed effects models with task stage, frequency and the interaction between task stage and frequency as fixed factors and participants as random intercept effect, using the maximum likelihood estimation. We performed simple slopes analyses to determine the region of significance of frequency for the fitted relationship between coherence and the task stages (where the 95% confidence interval did not include zero). The extracted point of maximum coherence (based on the defined confidence intervals) across task stages was correlated with ΔsAA using bootstrapped (n = 5000) repeated measures correlation.

For the fMRI analyses, we additionally correlated the extracted subject-specific mean LC-related beta-coefficients with emotional memory performance.

To determine which areas were lagging or leading in the spectral coherence analyses, we used the point of maximum coherence and extracted within the CIs the median coherence and phase lag per person. Two linear mixed models were fitted with median coherence or phase at maximum coherence as dependent variables and task stage, timeseries combination (LC, AMY, HIPP or EC) and the task stage by timeseries interaction as fixed effects and participants as random intercept factor. To examine that significant coherence between timeseries were independent of other timeseries of the network, partial coherences were calculated. For example, for three coherences between timeseries, we calculated the partial coherence in cases where significant coherence magnitude existed between region A – region B, region B and region C, and region A and region C and where the coherence phase of A < phase of B < phase of C. The spectral matrix between these timeseries was calculated and partial coherences within the CIs of median maximum coherence were extracted. The results were compared with Wilcoxon tests against a null distribution determined from the partial coherences of the matched-AR randomized timeseries. In addition, linear regressions were used to test associations between emotional memory performance and coherence at specific task stages per ROI pair. Coherence at baseline was entered as covariate.

Finally, within the structural LC analyses, we correlated the mean intensity of the LC from the MTR, MT-TFL, TFL and normalized TFL images to emotional memory performance. Similarly, Pearson correlations were also used to assess the relationship between the AES motion metric, normalized TFL mean intensity and emotional memory performance; and between LC intensity, ΔsAA, rMSSD and LC variance for each task stage.

All p-values were two-sided. Contrasts for interactions within the linear mixed effects models were computed using estimated marginal mean trends with reference grids and smooth functions and these results were corrected with the Tukey's test. Separate models, including correlations and regressions, were corrected for multiple comparisons using the False Discovery rate (FDR)-approach.

## Acknowledgements

We would like to thank Ms. Niky Vogt for her assistance in creating the experimental paradigm and recruiting participants.

This work is supported by a NWO VENI [451-14-035] to HILJ and supported by intramural support by the Faculty of Health Medicine and Life Sciences of Maastricht University to HILJ.

## Additional information

### Funding

| Funder | Grant reference number | Author |
|---|---|---|
| Nederlandse Organisatie voor Wetenschappelijk Onderzoek | VENI 451-14-035 | Heidi IL Jacobs |
| Universiteit Maastricht | Intramural support FHML | Heidi IL Jacobs |

The funders had no role in study design, data collection and interpretation, or the decision to submit the work for publication.

### Author contributions

Heidi IL Jacobs, Conceptualization, Data curation, Formal analysis, Supervision, Funding acquisition, Investigation, Methodology, Project administration, Data interpretation; Nikos Priovoulos, Data curation, Formal analysis, Investigation, Visualization, Methodology, Data interpretation; Benedikt A Poser, Software, Supervision, Data interpretation; Linda HG Pagen, Investigation; Dimo Ivanov, Software, Data interpretation; Frans RJ Verhey, Kâmil Uludağ, Data interpretation

### Author ORCIDs

Heidi IL Jacobs (iD) https://orcid.org/0000-0001-7620-3822

### Ethics

Human subjects: All participants provided written informed. Approval of the experimental protocol was obtained from the local ethical committee of the Faculty of Psychology and Neuroscience at Maastricht University (#07_11_2014_A1).

### Decision letter and Author response

Decision letter https://doi.org/10.7554/eLife.52059.sa1
Author response https://doi.org/10.7554/eLife.52059.sa2

## Additional files

### Supplementary files

• Supplementary file 1. Demographics and neuropsychological performance of the group.

• Supplementary file 2. Available data per modality and time point. Note: numbers provided are N and % compared to the total sample size (N = 27); HRV = heart rate variability

• Supplementary file 3. Task stages differences in sAA, rMSSD or LC BOLD variance. Note: Linear mixed effects models with random intercept for each person and task stage as fixed effect. Estimates indicate the unstandardized beta-coefficients. P-values are adjusted for multiple comparisons using the False Discovery rate.

• Supplementary file 4. Relationship between frequency and coherence between LC and heart rate variability across the task stages (Spatially normalized pipeline). Note: Linear mixed effects models with random intercept for each person, task stage, frequency and their interaction as fixed effect. Estimates indicate the unstandardized beta-coefficients. P-values are adjusted for multiple comparisons using the False Discovery rate.

• Supplementary file 5. Relationship between frequency and coherence between LC and heart rate variability across the task stages. Note: Linear mixed effects models with random intercept for each person, task stage, frequency and their interaction as fixed effect. Estimates indicate the unstandardized beta-coefficients. P-values are adjusted for multiple comparisons using the False Discovery rate.

• Supplementary file 6. Relationship between frequency and coherence between LC and heart rate variability across the task stages (Fixed + Explicit Resp pipeline). Note: Linear mixed effects models with random intercept for each person, task stage, frequency and their interaction as fixed effect. Estimates indicate the unstandardized beta-coefficients. P-values are adjusted for multiple comparisons using the False Discovery rate.

• Supplementary file 7. Relationship between frequency and coherence between LC and heart rate variability across the task stages (Fixed + Explicit Phys pipeline). Note: Linear mixed effects models with random intercept for each person, task stage, frequency and their interaction as fixed effect. Estimates indicate the unstandardized beta-coefficients. P-values are adjusted for multiple comparisons using the False Discovery rate.

• Supplementary file 8. Task-related activation patterns during encoding and recollection and functional connectivity (FC) alterations across the task stages. Note: Brain regions activated during encoding, recollection or functional connectivity differences across the conditions (Family-wise error corrected for multiple comparisons using TFCE at p<0.05). Coordinates are provided in 1 mm MNI-space.

• Transparent reporting form

### Data availability

Processed data to reproduce the figures in this manuscript is provided in the Source data files. Participants did not explicitly consent to their data being made public and therefore, access to their demographic, raw or processed imaging and physiological data is restricted. Requests for the anonymized data should be made to Heidi Jacobs (www.heidijacobs.nl; h.jacobs@maastrichtuniversity.nl or hjacobs@mgh.harvard.edu) and will be reviewed by an independent data access committee, taking into account the research proposal and intended use of the data. Requestors are required to sign a data sharing agreement to ensure participants' confidentiality is maintained prior to release of any data and that procedures conform with the EU legislation on the general data protection regulation and local ethical regulations.

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
