## [Decision Letter]

**Acceptance summary:**

This paper presents a rich study investigating Locus Coeruleus (LC) function with a variety of MR and physiological measures. The study of LC function has recently gained more interest due to the LCs early decline in dementia. The authors show that structural and functional measures of the LC can be used to gain insight in its role in memory encoding and retrieval as well as related physiological processes.

**Decision letter after peer review:**

Thank you for submitting your article "Dynamic behavior of the locus coeruleus during arousal-related memory processing: a multi-modal 7T fMRI study" for consideration by *eLife*. Your article has been reviewed by three peer reviewers, and the evaluation has been overseen by a Reviewing Editor and Timothy Behrens as the Senior Editor. The following individuals involved in review of your submission have agreed to reveal their identity: Mara Mather (Reviewer #3).

The reviewers have discussed the reviews with one another and the Reviewing Editor has drafted this decision to help you prepare a revised submission.

Summary:

This paper reports on an impressive study which presents the state of the art in the type and quality of measures needed for investigating Locus Coeruleus (LC) function, also including MR measures developed by the authors themselves. A shortcoming of the paper is that the analysis methods used are not always sufficiently documented to enable an assessment of their precision, usefulness and validity. Especially with a structure so rarely investigated in fMRI, a study with such a nice state-of-the-art set-up should help us understand what aspects of the LC are possible to investigate with fMRI and which aren't. In this light, I would also think that already a convincing report on basic task-related fMRI responses in the LC can present a sufficient scientific advance. More advanced additional analyses might not be needed if their physiological plausibility proves unsatisfying.

Essential revisions:

Specifically, the following revision requirements are suggested, also based on the comments by the external reviewers:

Introduction:

1) The authors should be clearer in the Introduction as to how LC/NE effects might differ between different stages of memory processing.

Behavioral analyses on memory measures:

2) The bias measure hasn't been sufficiently described to be understandable. Also given difficulties in the task design for the name recognition memory part, it is suggested to just focus on Hits-FAs of the emotional face recognition task (not the name recognition task).

Preprocessing of fMRI data:

3) The authors mention comparing different preprocessing pipelines but (to my knowledge) don't show statistics on how these result in differences in the GLM results or SNR measures in their regions of interest. These should be added and are in particular critical as the authors chose to proceed with a denoising approach that does not include respiration and pulse regressors (maybe outline to what extent these might be captured even without explicit regressors in the chosen ICA denoising approach).

4) Precision in the spatial coregistration/normalisation is paramount for group analyses on such a small structure. The authors should provide supplementary information that allows to evaluate the precision of the spatial transformations for every individual fMRI dataset.

Physiological/functional measures:

5) it isn't clear why a relationship between a change in sAA across subtasks and task-specific measures of rMSSD is more relevant than a relationship between task-specific sAA levels and rMSSD measures.

6) the significance of the coherence analyses linking LC BOLD variability and HRV/sAA are unclear (what does a higher coherence between HRV and BOLD variability mean? Why does this have to be related to a change in sAA?). Please only leave this in /a subset of analyses in) if the analyses can be reasonably physiologically motivated. Also, for these analyses, exclusively pulse-corrected fMRI data should be used.

7) Please scrutinize the GLM results for circularity in the ROI selection.

8) It is not clear what variability in the coherence measures between e.g. LC and MTL indicates functionally. It might be cautioned against interpreting these for areas/task conditions where the task-related GLMs did not show significant activations?

Discussion:

9) The Discussion should be scrutinized for claims that exceed the physiological significance of the results.

[Editors' note: further revisions were suggested prior to acceptance, as described below.]

Thank you for resubmitting your work entitled "Dynamic behavior of the locus coeruleus during arousal-related memory processing in a multi-modal 7T fMRI paradigm" for further consideration by *eLife*. Your revised article has been evaluated by Timothy Behrens (Senior Editor) and a Reviewing Editor.

The manuscript has been improved but there are some remaining issues that need to be addressed before acceptance, as outlined below:

We thank the authors for their work in revising this manuscript which has clarified several issues. There are still some concerns with the paper which we ask the authors to address:

1) As a general rule, we would like to ask the authors to go through the methods again and make sure that all analyses can be retraced by the reader. This concerns in particular the behavioral analyses. It is still not clear how the bias memory score was calculated. Please indicate which data were used for this (Hits-FAs of names related to which faces, old or new?) and give the precise formulas of all the calculations applied to these data.

2) Related to this, to add a more accessible measure of emotional memory effects (of names associated with faces), please add a table giving Hits, FAs, and Hits-FAs for names separately for old emotional and old neutral faces and separately for new emotional and new neutral new faces (incorrectly indicated as old (FA)), as well as statistics on the difference in Hit-FAs measures for names only on correct as old identified emotional versus neutral faces.

3) We ask the authors to consider limiting the analyses exploring links between LC resting state variability and (indirect) physiological indicators of NA (sAA and HRV) to correlations across task stages and analyses with HRF-convolved low and high HRV events, which are very interesting. By contrast, the coherence analyses in particular frequencies of resting state fMRI data with HRV seem not well motivated by the cited literature (which physiological processes do these high-frequency-specific fMRI data likely represent?). Alternatively, to taking these results out, the authors can revise their motivation for these analyses and make the interpretation of the results clearer. The reviewers found these hard to understand in an already complicated paper.

4) The significance of the exploratory coherence analyses, which report changes in the variability in coherence as their outcome measure seems to be also less clear in its physiological meaning (which processes that we can measure in fMRI are changing in coherence here and why should this be related to sAA?). Again, we invite the authors to remove these analyses. However, If the authors feel that this is adding valuable information, they are invited to leave these analyses in and make the interpretation of the results more accessible to the reader.

5) The results from the analyses on changes in coherence across task stages are not yet clear. I am wondering whether such analyses are prone to spurious results: if HIP-AMY coherence is higher in consolidation > baseline and in baseline > encoding, shouldn't it also be higher in consolidation > encoding? What does it mean if this effect is not picked up by the analyses? If the authors have easy access to the data, they may consider adding an analysis in a control area that shouldn't be implemented in consolidation or retrieval to provide cut-offs for avoiding spurious results or suggest alternative approaches to control for spurious results. Again, however, an alternative is to discuss this issue.

[Editors' note: further revisions were suggested prior to acceptance, as described below.]

Thank you for resubmitting your work entitled "Dynamic behavior of the locus coeruleus during arousal-related memory processing in a multi-modal 7T fMRI paradigm" for further consideration by *eLife*. Your revised article has been evaluated by Timothy Behrens (Senior Editor) and a Reviewing Editor.

The manuscript has been improved but there are some remaining issues that need to be addressed before acceptance, as outlined below:

We thank the authors for implementing most of the requested changes. However, in their responses to Comment 1 and 2, they misunderstood or overlooked important aspects of the question. We would like to ask them to adjust their responses and changes in the paper as follows:

Re Comment 1: Please also indicate whether the names used for the calculation of the bias score were paired with new or old faces or whether this was collapsed across both old and new faces.

Re Comment 2: There was a misunderstanding in the response to comment 2. The statistics required in the table were the hit and false alarm rate for *names*, not for *faces*. Please report the Hit rate, FA rate and Hit-FA rate separately for the names paired in the memory tests with old emotional and old neutral faces and for those paired with new emotional and new neutral new faces.

---

## [Author Response]

Essential revisions:Specifically, the following revision requirements are suggested, also based on the comments by the external reviewers:Introduction:1) The authors should be clearer in the Introduction as to how LC/NE effects might differ between different stages of memory processing.

We thank the reviewer for this excellent remark. Several animal studies have exactly tried to answer the question to what extent the LC plays a role in the different stages of memory processing. The animal literature is not consistent as different methods have been used (pharmacological manipulations, stimulation or lesion studies), and measurements were done at different time points along the task at hand.

With respect to encoding, studies injecting propranolol, a β-adrenoreceptor antagonist, prior to inhibition avoidance learning observed no effect on subsequent recognition tests [1-3]. While this may suggest that β-adrenoreceptors are not involved in learning, others have demonstrated that blocking β-adrenoreceptors prior to learning does affect long-term potentiation [4]. Possible explanations for these inconsistencies may be that activation of the LC induced release of dopamine [2], leading to (less strong) long-term potentiation. The timing and intensity of locus coeruleus activation may be critical here. The amplitude as well as the duration of norepinephrine release in the amygdala of rats after inhibitory avoidance training determined their subsequent memory enhancement effect [5].

It has been suggested that encoding of arousing information shifts norepinephrine neurons from tonic into phasic activity [6, 7]. This phasic signaling would sustain into and support consolidation. Studies infusing propranolol immediately after training in the basolateral amygdala demonstrated impaired recognition performance [1, 7-9], indicating that activation of the adrenergic system plays an important role in consolidation and neuronal plasticity. Interestingly, Khakpour-Taleghani and colleagues demonstrated that these latter effects may be time-dependent, as inactivation of the LC 90 or 360 minutes after learning had no effect on subsequent memory retrieval and suggests that later phases of consolidation may be dependent on distinct molecular processes.

Adrenergic signaling effects may also play a time-limited role in hippocampus-dependent memory retrieval. Stimulating the LC prior to the retrieval test enhanced [10] performance and inactivation of the LC 5 minutes before the retention test impaired memory performance. Retrieval could become independent of norepinephrine when the hippocampus is no longer playing a critical role in the consolidation process. This may explain why animals with very little norepinephrine in the forebrain are still able to perform well on retention tasks [11].

While outside the purpose of the current manuscript, it is important to note that other neuromodulators may affect memory formation by influencing norepinephrine neurons, such as for example glutamate, glucocorticoid, GABA, acetylcholine and estrogens.

In the Introduction:

“Research into NE-modulation of emotional memory enhancement has focused largely on the amygdala[6-8], an important target region of the LC and a critical nexus in memory for arousing experiences. Rodent work consistently demonstrated that blocking β-adrenergic receptors in the basolateral amygdala (BLA) immediately after learning in arousing conditions impairs memory, while infusion of NE or β-adrenoreceptor agonists enhances consolidation and rescinds the impairment, evaluated one to two days later [9, 10]. Correspondingly, stimulation of the LC during the learning phase modulates activity in the BLA and synaptic plasticity in the CA1, CA3 and dentate gyrus [11, 12]. While these studies highlight the importance of adrenergic signaling for plasticity and hippocampal-dependent learning, blocking β-adrenoreceptor prior to learning had no effect on subsequent retrieval 24 hours later [1, 2]. It has been suggested that in the context of strong LC stimulation tonic activity can be shifted into phasic activity and other neuromodulators may then elicit similar but possibly weaker plastic processes [4, 6, 7].”

In the Introduction:

“These network interactions can vary over time or be task-dependent. For example, the LC is involved in consolidation, but within a critical time period after learning, the time required for the hippocampus to facilitate the reorganization and stabilization of storing information in the neocortex [15].”

Behavioral analyses on memory measures:2) The bias measure hasn't been sufficiently described to be understandable. Also given difficulties in the task design for the name recognition memory part, it is suggested to just focus on Hits-FAs of the emotional face recognition task (not the name recognition task).

We apologize for not being clear in our methodology and explaining our setup. We will clarify this here and in the manuscript.

Concerning the bias measure: The bias measure originates from signal detection theory, often used in psychology [12]. The bias provides a metric that indicates the strategy of the participant, i.e. whether there is a bias towards a liberal answer (for example indicating that he or she saw the face previously) or a more conservative approach (for example answering that the face had not been seen previously). The advantage of using the likelihood ratio of the Beta is that it allows to interpret the values within a statistical framework. As Figure 2A-B (t=2.493, p<0.019) shows, there was a positive bias for the emotional faces compared to the neutral faces, hence subjects were more likely to indicate that they had seen the face earlier for the emotional condition than for the neutral condition. This bias will impact the retrieval scores, as participants will then receive more emotional trials during the retrieval condition relative to neutral trials. This is exactly what our data shows: the bias reduced retrieval performance (the participants falsely identified the faces as seen before) indicating that this was indeed a bias and not simply better recognition. This bias makes it hard to disentangle true recognition from false recognition, but the setup of our two-stage memory task and the application of signal detection theory, allow us to correct for this bias in the retrieval stage. Correcting for this bias cleaned up the retrieval scores (see Figure 2F in the manuscript). We believe that this will also be valuable to other studies with a similar approach.

Concerning our task design: We also would like to clarify the set-up of the task and the terminology that we used for differentiating between the memory processing stages. During encoding, participants view emotional and neutral faces coupled to a Dutch name. The encoding phase is followed by a resting-state period, which we term “consolidation”. Then, we conceptually distinguish recognition from retrieval. During recognition, we ask participants whether they believe they have seen the face previously (the face is shown without any names). If they endorse this question, they will be asked which name was associated to the face during the encoding phase. The latter is in our manuscript referred to as retrieval. Recognition or familiarity is a fast process that does not necessarily involve active retrieval of details surrounding the face [13]. We do recognize that the word “retrieval” may have been confusing, as in our task there is still no involvement of free or cued recall. Therefore, we have clarified in the manuscript that this process is more similar to recollection, the retrieval of qualitative information about a face-name pair. We should note that the participants had already been exposed to all names during encoding, so that this recollection step does not consist of simple name recognition. To avoid a bias in name recollection, all the incorrect (false positive) names during recollection were mixed in terms of valence.

Concerning the choice of recognition versus recollection: Recognition (familiarity) and recollection are two functionally independent processes and are also guided by two distinct brain systems. Familiarity is associated with lateral prefrontal and parietal regions, supporting processing associated with identifying the stimuli. On the other hand, recollection involves medial prefrontal, medio-lateral parietal and hippocampal regions. The hippocampus has been related to familiarity processes under high confidence ratings, such as recollection, given that recollection relies on mechanisms related to pattern separation and pattern completion [13-16]. See also the below additional fMRI GLM analyses for recognition. As expected, we do see activity in the fusiform gyrus (successful recognition) and lateral prefrontal regions (successful emotional recognition), but not for the hippocampus.

Data of the recognition part of the task: Following the advice of the reviewer, we examined the hit and false alarm rates of the face recognition part of the task. There was no significant difference (Wilcoxon paired test: Z = -1.24, p-value=0.222) between the recognition hit rate for emotional (Median = 53.33% (IQR=45.56 – 61.11)) vs neutral (Median = 53.33% (IQR = 44.44-67.79); Figure 1A). In accordance with our report of a bias during face-recognition, a significant increase of false alarms was detected (Wilcoxon paired test: Z=3.86, p-value<0.001) for emotional (Median = 22.73% (IQR = 18.19-34.09) compared to neutral faces (Median = 13.63% (IQR = 9.09- 22.78); Figure 2B).

Brain activation patterns related to recognition: A general linear model was fit at each voxel with FMRI Expert Analysis Tool (FEAT), using generalized least squares with a voxel-wise, temporally and spatially regularized autocorrelation model over the whole brain. A high-pass filter (cutoff at 100 s) and minimal spatial smoothing (kernel = 1.5 mm) were applied before the fit. The regressors consisted of the emotional trials that were later successfully recognized, the neutral trials that were later successfully recognized, the emotional trials that were not recognized and the neutral trials that were not recognized. Contrasts of interests were emotional trials vs. neutral trials (Emotional > Neutral), successfully recognized trials vs. not successfully recognized (Successful recognition > Not successful recognition) and the interaction (Emotional successful recognition > neutral successful recognition).

Significant activation following cluster correction was detected for the (Emotional > Neutral) contrast in the right middle temporal gyrus (peak z-value = 3.77, MNI coordinates [68,-35,-5], cluster volume = 157 mm; Figure 6—figure supplement 3). For the (Successful recognition > Not successful recognition) contrast significant activation was found in the right temporal fusiform cortex (peak z-value = 4.13, MNI coordinates [39,-35,-21], cluster volume = 101 mm). For the interaction (Emotional successful recognition > neutral successful recognition) a significant cluster was detected in the right inferior frontal gyrus (peak z-value = 4.08, MNI coordinates [57,29,12], cluster volume = 191 mm). No clusters were visible in the amygdala, hippocampus, entorhinal cortex or locus coeruleus for the above contrasts.

In the Results:

“This was followed with a recollection fMRI-scan, during which participants had to indicate whether they recognized the face and, upon endorsement, had to select the name of the face out of three options (all options were seen previously to stimulate recollection instead of recognition processes).”

In the Results:

“There was no significant difference (Wilcoxon paired test: Z = -1.24, p-value=0.222) between the recognition hit rate for emotional (Median = 53.33% (IQR=45.56 – 61.11)) compared to neutral faces (Median = 53.33% (IQR = 44.44-67.79). However, more false alarms were detected (Wilcoxon paired test: Z=3.86, p-value<0.001) for emotional (Median = 22.73% (IQR = 18.19-34.09) compared to neutral faces (Median = 13.63% (IQR = 9.09- 22.78); Figure 2B).”

In the Results:

“We decided to focus on recollection because of this response bias, our focus on the medial temporal lobe regions and the notion that recollection is more likely to elicit hippocampal activity than recognition.”

In the Materials and methods:

“This response bias was derived from the signal detection theory, a method that allows us to differentiate the signal from the noise. The bias measure indicates the extent to which a certain type of response is more probable than another, such as for example responding in a more conservative manner. The bias was regressed out from the emotional and neutral recollected scores and using their residuals this resulted in an adjusted emotional memory performance score.”

In the Materials and methods:

“Contrasts related to recognition were not our primary interest, however, we fit encoding regressors for emotional trials that were later successfully recognized, the neutral trials that were later successfully recognized, the emotional trials that were not recognized and the neutral trials that were not recognized. Contrasts of interests were emotional trials vs. neutral trials (Emotional > Neutral), successfully recognized trials vs. not successfully recognized (Successful recognition > Not successful recognition) and the interaction (Emotional successful recognition > neutral successful recognition). Results are presented in Figure 6—figure supplement 3.”

Figure 2 was updated. We also edited all the figures, tables, supplementary data and text replacing retrieval by recollection to avoid any confusion among the readers. We added the fMRI results for the recognition to the supplemental data. We also update the data file for this figure.

Preprocessing of fMRI data:3) The authors mention comparing different preprocessing pipelines but (to my knowledge) don't show statistics on how these result in differences in the GLM results or SNR measures in their regions of interest. These should be added and are in particular critical as the authors chose to proceed with a denoising approach that does not include respiration and pulse regressors (maybe outline to what extent these might be captured even without explicit regressors in the chosen ICA denoising approach).

We agree with the reviewer that this is a crucial point for fMRI experiments in general and brainstem fMRI in particular. Overall, as described in our methods the approach we followed to mitigate the chance that our results are due to unmodeled physiological noise was to a) repeat the analysis across pipelines and b) repeat the analysis with a reference component that is adjacent to the LC. We further collected respiration and heart-timeseries and included them in a denoising scheme (using FSL’s PNM) along with a model-free physiological denoising based on FSL-FIX. Crucially, this was combined within a single regression to ensure that noise is not reintroduced in the data. Overall, when we examined the impact of our denoising pipelines on tSNR in the baseline resting state (Figure 2) we found that including FIX provided a sizable gain in tSNR, while including explicit physiological time series provided a more modest tSNR increase (median FIXed=39.78 (IQR = 35.862 – 42.88), median FIXed +Explicit Resp = 44.74 (IQR = 39.893 – 51.931), median FIXed +Explicit Phys = 46.39 (IQR = 41.402 – 54.159), median Spatially normalized = 23.48 (IQR = 21.657 – 26.156)), but at the cost of removing 22 statistical degrees of freedom (while the number of volumes where only 180 e.g. in the resting stage scans). This is also reflected in the intraclass correlation coefficient between the pipelines: Spatially normalized versus FIXed: ICC=0.443; FIXed versus FIXed +Explicit Resp: ICC=0.96; and FIXed and FIXed +Explicit Resp and FIXed +Explicit Phys: ICC=1.00. These ICC values show a high level of similarity between the tSNR of the FIXed, FIXed +Explicit Resp and FIXed +Explicit Phys pipelines. Based on the above, we were not convinced that the explicit physiological regression provided a robust improvement over the implicit one (FIXed) (especially when considering that regressing-out even not very-meaningful regressors tends to reduce variance and thus increase the tSNR) and we chose to focus our report in the FIXed dataset.

The reason that the gain from the inclusion of physiological regressors is modest is that a) FSL-FIX already regresses out physiological-noise components b) respiration and heart-time series tend to get corrupted periods over lengthy scan sessions. Even so, we repeated the analysis across the pipelines to ensure the robustness of our results. Overall results between FIXed and the FIXed +Explicit Resp/Phys were largely consistent. Note that we changed the names of our pipelines as suggested by the reviewers in comment 18. See Figure 1—figure supplement 2, Figure 4—figure supplement 1, Figure 4—figure supplement 2 and Figure 5—figure supplement 1 for details.

LC variability - rMSSD relationship across preprocessing steps:We performed the repeated measures correlations between LC BOLD variability and rMSSD across all preprocessing steps. Comparisons across pipelines show similar effects in the latter two pipelines (Figure 4—figure supplement 2B and C) as in the FIXed pipeline reported in our manuscript (Rrm=-0.387; p = 0.018).Locus coeruleus - HRV coherence during task across preprocessing steps:We also performed the coherence analysis between LC and HRV across all preprocessing steps.

Spatially normalized:

We ran linear mixed effect models to examine whether the relationship between frequency and coherence between the LC and heart rate variability varied across task stages. See supplementary file 4 and Figure 5—figure supplement 2.

FIXed:

We also ran linear mixed effect models to examine whether the relationship between frequency and coherence between the LC and heart rate variability varied across task stages. See Supplementary file 5 and Figure 5—figure supplement 3.

FIXed +Explicit Resp:

For the FIXed +Explicit Resp, we also ran linear mixed effect models to examine whether the relationship between frequency and coherence between the LC and heart rate variability varied across task stages. See Supplementary file 6 and Figure 5.

FIXed +Explicit Phys:

For the FIXed +Explicit Phys, we ran examined whether the relationship between frequency and coherence between the LC and heart rate variability varied across task stages using linear mixed effects models. See Supplementary file 7.

Overall, checking the reproducibility of our results showed that across all preprocessing pipelines – except the spatially normalized pipeline – the coherence between LC and HRV is greater during baseline compared to consolidation, during recollection compared to baseline, during recollection compared to encoding and during encoding compared to consolidation. These results are similar to the results reported in the original manuscript for the FIXed-pipeline.

Functional connectivity (FC) from the LC component to MTL structures across task stages.

We performed seed-to-voxel analyses to explore the cross-correlation at lag = 0 s (denoted as FC) between the LC (consisting of the LC component) and medial temporal lobe structures (set in advance to include hippocampus (HIP), amygdala (AMY) and entorhinal cortex (EC)). This revealed a significant decrease of FC between LC and left BLA, subiculum and entorhinal cortex across preprocessing pipelines, except for the spatially normalized data where no significant cluster-wise corrected changes in FC were detected. See Supplementary file 8 and Figure 7—figure supplement 1.

In the Results:

“To assess the specificity of our findings to the LC, we also investigated a control component in the pons (reference ROI). In addition, sensitivity analyses were done by evaluating the outcomes across four different preprocessing pipelines, each time adding in more physiological noise correction (Figure 4—figure supplement 1).”

In the supplemental data:

We added the results of each pipeline to the supplemental data, as well as the tSNR comparison across the pipelines. Following comment 6 of the reviewers, we used the FIX+PNM pipeline for the coherence analyses (instead of the FIX only pipeline).

The data file associated with this figure was updated.

4) Precision in the spatial coregistration/normalisation is paramount for group analyses on such a small structure. The authors should provide supplementary information that allows to evaluate the precision of the spatial transformations for every individual fMRI dataset.

We are happy to provide more information on the spatial coregistration. For the fMRI scans, it was crucial that quality registrations to the MNI template were obtained to facilitate integration with previous results and existing atlases. This is challenging with gradient-echo scans when aiming to image such a small region at high resolution and given the required coverage and SNR for memory experiments. Gradient-echo scans are inherently sensitive to dephasing and thus distortion in the phase-encoding direction. To tackle this, we combined careful placement of the field of view (so that a) the quickly-dephasing sinuses in the frontal and temporal lobes are largely excluded and b) the phase-encoding gradients are applied orthogonally to the main axis of the hippocampus and 45 degrees to the main axis of locus coeruleus so that their displacement is minimized) with parallel MRI (so that the slice thickness and thus the inclusion of quickly-dephasing regions within the relevant slices would be reduced). Having acquired scans that showed reduced distortions in the relevant ROIs, we calculated a warp field to minimize distortion in the rest of the brain and facilitate registration using FSL TOPUP.

The T1-weighted scans were preprocessed prior to the Freesurfer pipeline and subsequent registrations: typical MP2RAGE scans, while high-resolution and theoretically bias-corrected, show background noise outside the brain. To minimize this, the scans were recombined following the process described in [17] so that the background noise was suppressed. Then the scans were bias field corrected for remaining inhomogeneities using an N4 algorithm and brain extraction was performed. Freesurfer was performed using our own brain mask and both the T1-weighted image and T1-map (that was used as a substitute for a T2-weighted image) to improve white matter and grey matter delineation. The Freesurfer segmentations were then used as input to estimate a boundary-based registration transform from the fMRI images to the T1-weighted images using FSL. Subsequently, the N4-corrected T1-weighted scans were used to calculate diffeomorphic transforms to the MNI template with ANTs (symmetric normalization transformation model; SyN; antsRegistrationSyN.sh). The SyN algorithm has been shown to provide superior performance compared to most other registration algorithms used in neuroimaging [18, 19]. The linear transformations were transformed to the ITK-SNAP reference using c3d tools and combined with the warps, so that when transforming the data from native fMRI to MNI there was a single interpolation. The accuracy of the transforms was individually checked by NP.

To allow reviewers to examine the registrations, we are showing the mid-sagittal, coronal and axial slice of the MNI-warped T1-weighted and fMRI scans for every individual (Author response images 1-4). The quality of the registrations can be visually assessed. Furthermore, we created an average of all the individual T1-weighted and fMRI scans warped to the MNI space (Figure 1—figure supplement 1). The fact that small but relevant features of the hippocampus and brainstem can be clearly observed in both averages support our co-localization across group locus coeruleus and medial temporal lobe activation. Finally, we attempted to provide a quantitative metric of the quality of registrations between individuals. Typically, this is done with metrics of volume overlap or distance in previously-labelled datasets [19]. This is labour intensive to do manually, so we opted to use the automatic labelling of the Freesurfer pipeline. We created individual ROIs consisting of the Freesurfer’s segmentation of hippocampus and brainstem, as the most relevant anatomical features [20, 21]. The ROIs were projected to the MNI space. Then for each individual ROI A, we calculated a measure of volumetric overlap versus every other volumetric ROI B, such as overlap ratio = ((A)U(B))/(A). The distribution of the overlap ratio per individual is shown in Author response image 5. The overlap ratio across the group was high (median=0.93) with a very tight spread (IQR=[0.92-0.94]) indicating even registration performance across the group. We should note that this process likely underestimates the registration quality, since it relies on the automated segmentation of Freesurfer.

Finally, we would like to draw the attention of the reviewers to Figure 8 in the manuscript, where the locus coeruleus MT-weighted template is shown. Note that despite the very small size of the locus coeruleus (2mm wide), it can easily and specifically be discerned at group level, further attesting to our capacity to image the nucleus across group.

**Author response image 1. sa2fig1:** Mid-sagittal, coronal and axial slices in the MNI-space for every T1-weighted (first 3 columns) and fMRI-dataset (last 3 columns). Every row represents a participant. The first row shows the relevant MNI slices. Also see Author responses image 2-4.

**Author response image 3. sa2fig3:** 

**Author response image 4. sa2fig4:** 

**Author response image 5. sa2fig5:** Overlap ratio of individual ROI (Freesurfer-segmented hippocampus and brainstem) versus the matched ROI across the group. The distribution for each individual is shown in the x-axis.

In the Materials and methods:

“The T1 to MNI template registrations were calculated using a diffeomorphic transform with the antsRegistrationSyN.sh function (ANTS 2.1 (http://stnava.github.io/ANTs/) [18]). This algorithm has been shown to provide superior performance compared to most other registration algorithms used in neuroimaging [18, 19].”

In the Materials and methods:

“A boundary-based registration was calculated for the linear registration between the fMRI and T1data using FSL's epi_reg based on the FreeSurfer segmentations. The T1 to MNI template registrations were done using ANTS as described in the preprocessing of the structural data. The linear transformations were transformed to the ITK-SNAP reference using c3d tools and combined with the warps, so that when transforming the data from native fMRI to MNI there was a single interpolation. The accuracy of the transforms was individually checked by NP.”

Given that *ELife* also publishes the response letter, we decided to not overwhelm the reader with supplemental data. If this paper would be accepted for publication, the reader will be able review the individual images in this response letter. We added the T1 and fMRI average image to the supplemental data and made reference to the individual images in the response letter.

Physiological/functional measures:5) it isn't clear why a relationship between a change in sAA across subtasks and task-specific measures of rMSSD is more relevant than a relationship between task-specific sAA levels and rMSSD measures.

We performed both: we examined changes in sAA and rMSSD across the different stages of the memory task using linear mixed effects models (Figure 3A and 3B in the manuscript) and we examined person-specific relationships between two measures along the entire setup using repeated measures correlations (Figure 3D in the manuscript). We don’t think that one set of analyses is more relevant than the other, but that they provide complementary information. The repeated measures correlation is similar to a linear mixed effects model, but allows for a time-varying predictor and time-varying outcome, with each having the same amount of measurements. Thus, in the first set of analyses, we examined whether one metric changes during specific parts of memory processing, whereas in the second set we explore whether the changes in one measure go together with changes in the other. Based on the physiology of the autonomic system and the homeostatic principles of the brain, we assume that sAA and rMSSD covary consistently across stages. We explained this in the Results section.

We like to point out that to enable the comparison of both measures along similar time points, we subtracted the sAA measures pre and post the condition (see Figure 1). For the rMSSD we calculated the variability during that time frame. Our data indicates that sAA during consolidation (note: this represents the differences between post-encoding versus post-consolidation) was elevated as compared to baseline (difference between pre-baseline and post-baseline). We did not see differences in rMSSD across the task stages. The fact that both sAA and rMSSD are correlated across the stages suggests that they are both measuring a similar biological construct and provides extra validity to our measures.

In the Results section:

“To facilitate analyzing rMSSD and sAA at similar time points along the paradigm, we calculated the difference in sAA (ΔsAA) levels before and after each of the stages of the fMRI task …”

In the Results section:

“Given that both measures relate to the autonomic system, we then correlated ΔsAA to RMSSD across the stages using bootstrapped (n=5000) repeated measures correlations”.

6) the significance of the coherence analyses linking LC BOLD variability and HRV/sAA are unclear (what does a higher coherence between HRV and BOLD variability mean? Why does this have to be related to a change in sAA?). Please only leave this in /a subset of analyses in) if the analyses can be reasonably physiologically motivated. Also, for these analyses, exclusively pulse-corrected fMRI data should be used.

We thank the reviewers for the opportunity to further demonstrate the importance of these analyses. The locus coeruleus neurons mediate arousal and autonomic activity. Autonomic activity is an intrinsic part of arousal or emotional experiences during a task. We already observed an overall correlation between locus coeruleus BOLD variability and HRV across the task stages (Figure 4 in the manuscript). But by combining our physiological data in a frequency-resolved manner with the LC BOLD data, we were able to examine the nature and contributions of autonomic regulations on LC neural activity across the different task stages.

The low frequency component (LF, 0.05–0.15Hz) is thought to be influenced by, though not entirely, sympathetic activity, while the high frequency component (HF, 0.15–0.40Hz) is influenced by parasympathetic, vagally mediated activity. Recent work demonstrated a frequency-specific relationship between the HF-HRV and brainstem activity [22, 23], which was interpreted to reflect the feedback loop between brainstem nuclei and parasympathetic system, which is expected to have a specific frequency profile (i.e. it is reflected in the high-frequency of HRV).

The coherence measure between HRV and LC BOLD is a measure of the magnitude of coupling between LC BOLD and HRV measures at specific frequencies.

The greater coherence between LC BOLD and HRV during recollection compared to baseline or encoding, indicates that as LC activity increases, the inhibition on the heart increases. This may reflect the feedback loop between cortical regions and the LC allowing for bidirectional communication between central memory areas and peripheral sites. This action allows conscious experience of the stimuli through top-down inhibitory influences on the LC, which shapes our arousal experiences and ultimately facilitates the appropriate response. Higher HF-HRV has been associated with better discrimination between emotional versus neutral faces, attentional resources and better recollection of context information from stored representation to perform memory task successfully [24, 25]. On the other hand, consolidation demonstrated lower coherence compared to encoding or baseline, which is consistent with the idea that lower HF-HRV is associated with higher arousal. The maximum coherence measure – mainly reflecting parasympathetic contributions – was negatively related to sAA, demonstrating that higher arousal-related metrics (lower coherence) is associated with greater sympathetic activity. The fact that our previous analyses demonstrated that sAA increased during consolidation and that this sAA increase related to memory performance confirms that consolidation is more likely to be influenced by sympathetic activity than parasympathetic activity. Such sympathetic influences may potentially reflect norepinephrine-related long-term potentiation.

Our sensitivity analyses, i.e. reproducing the same results using different processing pipelines highlight the robustness of these findings and exclude confounding by respiration.

We now clarified to the reader the rationale as well as the interpretation of these analyses in the Results section. We also added the results of the other pipelines to the supplementary data.

In the Results section::

“In the previous analyses we demonstrated a negative correlation between LC variability and rMSSD or ΔsAA across the entire paradigm. These results echo previous work demonstrating that blocking β-adrenoreceptor increases HRV [26]. However, autonomic fluctuations can alter the peak of frequencies in the HRV leading to complex patterns of variability. To understand the nature of autonomic regulations on LC activity at each task stage, we applied a time-frequency analysis by calculating the coherence magnitude squared and phase lag between the timeseries of the HRV and LC for each task stage (see example in Figure 5A). By using a within-subject design with a baseline measure we can eliminate interindividual differences in blood pressure, respiration and, by combining HRV fluctuations with other measures related to the autonomic system, we are well-positioned to differentiate autonomic contributions to LC activity due to emotion or arousal across the different memory stages. Here, we will use the FIXed+explicit Phys pipeline to remove respiratory confounding.”

In the Results section:

“These patterns indicate greater LC activity coupled to greater parasympathetic inhibition during baseline and recollection, which has been associated with better discrimination between emotional versus neutral faces, attentional resources and better recollection of context information from stored representation to perform memory task successfully. On the other hand, consolidation demonstrated lower coherence compared to encoding or baseline, which is consistent with the premise that lower HF-HRV is associated with higher arousal.”

In the Results section:

“The covariation between LC activity and HRV demonstrate that even though both emotion and cognition operate simultaneously, encoding and recollection are coupled to autonomic regulation supporting cognitive functions required for the task at hand. However, this LC-parasympathetic coupling is not sufficient to predict emotional memory performance. Conversely, LC BOLD signal during consolidation may be specifically tied to arousal-related sympathetic tone, an inference supported by our ΔsAA analyses and the at-trend correlation between ΔsAA during consolidation and memory performance.”

Figure 5: was updated to show the data with the FIXed+Explicit Phys pipeline

Supplemental data We added the results for these analyses with the other pipelines to the supplemental data.

7) Please scrutinize the GLM results for circularity in the ROI selection.

The ICA component of the locus coeruleus was identified in a separate resting-state period, that preceded the task (see Figure 4 —figure supplement 1 or Figure 4 in this response letter to comment 3). For the GLM results, we created a mask covering all regions of interest (the entire pons, medial temporal lobe as well as the 4^th^ ventricle – the latter to allow for detection of motion related fMRI signal). The ICA component was used in the analyses leading up to Figure 4 and 5 but were not used in the GLM analyses. All GLM analyses were done voxel-based and hence, there was no circularity.

We added a figure to the supplement demonstrating the mask with the regions of interest for the GLM and functional connectivity analyses.

8) It is not clear what variability in the coherence measures between e.g. LC and MTL indicates functionally. It might be cautioned against interpreting these for areas/task conditions where the task-related GLMs did not show significant activations?

Coherence is a mathematical technique that quantifies the frequency and amplitude of the synchronicity of brain activity in different regions. It estimates the consistency of the amplitude and phase between both BOLD signals over time within a specific frequency band. The amplitude provides information on the strength of the coupling or connectivity between two regions. The phase provides information on their temporal relationship. These analyses are important as arousal and salience processing in the brain, typically involving contributions of the LC, exhibit variable behavior, as we demonstrate in the relationship between LC and the HRV (comment 6). In static fMRI analyses, such as for typical functional connectivity (Figure 7A in the manuscript), this is ignored, rendering them hard to interpret, given that they can dilute specific sources of information. However, the brain can show dynamic changes not only within the frequency domain but over time as well. Previous research suggested that attention-related variability of connectivity in the time-frequency domain explains anti-correlated functional connectivity [27]. The wavelet coherence is a method to disentangle differences in the strength of the coupling between regions across task stages in a time-resolved manner. In our study, we found anti-correlated functional connectivity between the LC and the MTL. Based on the coherence results, we suggest that this finding potentially relates to dynamic functional connectivity between the LC and the MTL, as demonstrated by the variability in the wavelet coherence measure (i.e. this reflects the extent to which the coherence between regions can transition across time at a given frequency).

This analysis provides complementary information to the static functional connectivity analyses and therefore, we decided to examine all regions of interest and applied an adjustment for multiple comparisons. We clarified the premise for these analyses and the interpretation in the Results section.

In the Results section:

“Areas related to arousal and salience processing, such as the LC, have particularly variable behavior that can result in a negative FC value, complicating their interpretation [28].”

In the Results section:

“Integrating these findings with our LC-HRV coherence analyses (Figure 5) reveals correspondence for the increase in coherence variability during consolidation for all LC-MTL pairs and suggests that these time-varying behaviors in the dynamics between regions reflect changes in arousal and possibly sympathetic tone.”

Discussion:9) The Discussion should be scrutinized for claims that exceed the physiological significance of the results.

We ensured that our Discussion is reflective of our findings and whenever we speculated on the interpretations, we made that clear as well.

[Editors' note: further revisions were suggested prior to acceptance, as described below.]

We thank the authors for their work in revising this manuscript which has clarified several issues. There are still some concerns with the paper which we ask the authors to address:1) As a general rule, we would like to ask the authors to go through the methods again and make sure that all analyses can be retraced by the reader. This concerns in particular the behavioral analyses. It is still not clear how the bias memory score was calculated. Please indicate which data were used for this (Hits-FAs of names related to which faces, old or new?) and give the precise formulas of all the calculations applied to these data.

We are happy to provide more information on our methods. We went through our Materials and methods section and added more information. As specifically requested by the reviewers, we have also provided the detailed methods relating to the bias scores and which data was used. To make this clearer, we have now also added a figure that explains the procedure to calculate the adjusted emotional memory performance score.

In the Materials and methods section:

“We calculated hits and false alarms for both the emotional and neutral condition during encoding. For the recollection (identification of the name), we calculated hit and false alarm rates for both the neutral and emotional condition. Reaction times were also analyzed. Performance in the recollection condition may be affected by a response bias in the recognition phase between emotional and neutral stimuli. Therefore, we calculated the likelihood ratio beta, as a measure of the response bias (beta = exp (d' x c)) (d' index calculated as by Stanislaw and Todorov, but corrected for ceiling effects [29]). This response bias was derived from the signal detection theory, a method that allows us to differentiate the signal from the noise [12]. The bias measure indicates the extent to which a certain type of response is more probable than another, such as for example responding in a more conservative manner.

We performed a paired t-test to compare the bias between emotional and neutral during the recognition phase. This bias was then regressed our from the emotional and neutral recollected hits in two separate regressions (Figure 2). The residuals of each regression represented the bias-corrected emotional and neutral recollection rate. Finally, to obtain an individual emotional recollection rate, we calculated an adjusted emotional memory performance score = (Emotional recollection rate)/(Emotional recollection rate+Neutral recollection rate) (Figure 2—figure supplement 1). This measure was used throughout the manuscript.”

2) Related to this, to add a more accessible measure of emotional memory effects (of names associated with faces), please add a table giving Hits, FAs, and Hits-FAs for names separately for old emotional and old neutral faces and separately for new emotional and new neutral new faces (incorrectly indicated as old (FA)), as well as statistics on the difference in Hit-FAs measures for names only on correct as old identified emotional versus neutral faces.

This is an excellent suggestion and we have now also added in a Table in the Supplementary files with the median and IQR for Hits, False Alarms, Hit-False Alarms, Bias scores and adjusted memory scores for each condition and provided statistical comparisons between emotional and neutral scores. The table contains the medians and interquartile range for the recognition hit rate for old faces only, the correct rejection rate of new faces during recognition, the false alarm rate during recognition (i.e. incorrectly identifying new faces as old) and the difference between the recognition hit rate and the false alarm. Furthermore, we also included the raw recollection hit rate of the raw data (i.e. correct selection of name over the total number of old faces), the recollection hit rate following correct recognition and the recollection hit rate corrected for the recognition response bias. The normalized emotional the recollection hit rate corrected for the recognition response bias was used for all analyses in the rest of the manuscript). While making the table, we also noticed we had reported the Wilcoxon’s W statistic instead of the z-value for the bias-corrected recollection differences. For consistency, we now report the z-value for all comparisons.

In the Results:

“…participants recollected on average more names with an emotional (median=38.68, IQR=36.68, 39.46) relative to those with a neutral valence (median=29.97, IQR=28.01, 33.39; paired Wilcoxon test: Z=4.41, p-value<0.001, Figure 2G),…”

3) We ask the authors to consider limiting the analyses exploring links between LC resting state variability and (indirect) physiological indicators of NA (sAA and HRV) to correlations across task stages and analyses with HRF-convolved low and high HRV events, which are very interesting. By contrast, the coherence analyses in particular frequencies of resting state fMRI data with HRV seem not well motivated by the cited literature (which physiological processes do these high-frequency-specific fMRI data likely represent?). Alternatively, to taking these results out, the authors can revise their motivation for these analyses and make the interpretation of the results clearer. The reviewers found these hard to understand in an already complicated paper.

We thank the reviewers and editors for an opportunity to clarify the importance of these analyses. Within this manuscript we aimed to examine the role of the LC during arousal-related memory across the various stages. We found a negative correlation across task stages between sAA and rMSSD, a metric of heart rate variability (HRV). The rMSSD is a frequency non-specific measure, though the parasympathetic modulation of HRV is known to be frequency-specific (0.15 to 0.4Hz). We therefore explored which HRV frequency band has the largest effect on the LC fMRI signal. Remarkably, we found that the confidence interval of the individual maximum coherence between LC and HRV at baseline overlaps with the known frequency band of parasympathetic modulation of HRV. Furthermore, coherence within that frequency band negatively correlated with sAA levels. Given that the LC plays a crucial role in both autonomic processes and arousal, these coherence analyses help us to disentangle contributions of emotion or arousal to LC activity during the various stages.

Using the coherence method is attractive because it provided us with a single-point metric of LC-related autonomic modulation across memory task stages. The power of the low and high frequencies are not fixed and can vary in response to autonomic changes [26, 30]. This analyses was motivated by the fact that higher arousal due to emotionally salient stimuli can shift the low and high frequency states of the HRV states and occurs in concert with changes in BOLD responses in brain regions involved in emotional processing [31, 24]. Regarding the high-frequency of LC fMRI signal, we were primarily focused on the frequency-specificity of the HRV and while brainstem regions seem to show more connectivity at higher frequency as compared to the often lower-frequency networks observed in the cortex, we agree that the real physiological relevance of high-frequency fMRI is still unclear.

We have clarified the rationale of these analyses in the text, provided additional literature references, rephrased the interpretation, and have also briefly discussed the fact that the neurophysiological basis of high-frequency fMRI signals remains unclear.

In the Results:

“However, rMSSD reflects beat-to-beat variance in HR in the time-domain and is relatively non-specific to the frequency-domain. Autonomic fluctuations due to emotionally salient stimuli can alter the power of frequencies in the HRV and this can occur in concert with changes in LC BOLD responses [26, 30, 24]. Therefore, we applied spectral analyses to assess the nature of autonomic modulations on LC activity at each task stage by calculating the coherence magnitude squared between the timeseries of the HRV and LC for each task stage (see example in Figure 5A).”

“The covariation between LC activity and HRV demonstrate that even though both emotion and cognition operate simultaneously, encoding and recollection are coupled to moment-to-moment autonomic regulation supporting cognitive functions required for the task at hand. However, this LC-parasympathetic coupling, appertaining to the central autonomic network, is not sufficient to predict accurate emotional memory recollection.”

“We should note that the physiological relevance of high-frequency fMRI signal remains unclear, but it has indeed been linked to rapid modulation of the brainstem driving information exchange [32].”

4) The significance of the exploratory coherence analyses, which report changes in the variability in coherence as their outcome measure seems to be also less clear in its physiological meaning (which processes that we can measure in fMRI are changing in coherence here and why should this be related to sAA?). Again, we invite the authors to remove these analyses. However, If the authors feel that this is adding valuable information, they are invited to leave these analyses in and make the interpretation of the results more accessible to the reader.

We understand the reviewers concern. Two different methods measuring coherence can be confusing to the reader and therefore we decided to remove the wavelet coherence variability analysis. Thus, throughout the manuscript, the same spectral coherence method is now used: first, to examine interactions between the HRV frequency band and the LC fMRI signal, and second, as an exploratory analysis relating to negative stationary FC between the LC and MTL regions.

The goal of both the wavelet analyses and the spectral coherence analyses was to further understand these negative functional connectivity patterns. Changes in arousal, mediated by the ascending arousal system, are associated with increases and decreases in the fMRI signal [33]. For example, the amplitude and extent of correlations in fMRI data vary with indicators of drowsiness and light sleep [34-37], are altered by sleep deprivation [38] and reveal changes due to caffeine-induced fluctuations in arousal [39].Thus, these variations in arousal can induce phase-differences or frequency-specific magnitude correlations between regions. This may result in negative correlations in stationary functional connectivity analyses, as also shown in our own functional connectivity analyses across the different task stages (Figure 7). Phase differences between states may vary over time, as a function of arousal [40]. Neuromodulators, such as noradrenalin modulate the activity of a neuron and consequently, coherent neural communications are based on the dynamics of neurotransmitters. In fact, pharmacological manipulation of catecholamines has led to alterations in non-stationary functional connectivity [41].

Spectral coherence analysis allows the detection of magnitude and phase correlations at frequency specific levels between timeseries and is therefore well-suited to investigate frequency-specific phase and magnitude changes between task stages. In support of our hypothesis, we found magnitude correlation changes between the HIP-AMY between task stages and phase modulations between the AMY-LC. Coherence between the MTL regions was also shown to relate to emotional memory performance. We have removed the wavelet coherence analyses (removed the methods, associated statistical analyses, results and Figure 7—figure supplement 2) and rewritten this section.

In the Results:

“Areas related to arousal and salience processing, such as the LC, have particularly variable behavior that can result in a negative FC value, complicating their interpretation [27]. As our FC analyses assume stationarity within the fMRI timeseries, we aimed to understand these negative FC changes by exploring the frequency-specific magnitude and phase correlation changes in the follow-up spectral coherence analyses. Phase or magnitude changes in the frequency plane can result in reduced cross-correlation values.”

5) The results from the analyses on changes in coherence across task stages are not yet clear. I am wondering whether such analyses are prone to spurious results: if HIP-AMY coherence is higher in consolidation > baseline and in baseline > encoding, shouldn't it also be higher in consolidation > encoding? What does it mean if this effect is not picked up by the analyses? If the authors have easy access to the data, they may consider adding an analysis in a control area that shouldn't be implemented in consolidation or retrieval to provide cut-offs for avoiding spurious results or suggest alternative approaches to control for spurious results. Again, however, an alternative is to discuss this issue.

The reviewer is indeed right: we inadvertently changed the order of the task stages in the results text. We found increased coherence in encoding > baseline and decreased coherence in Recollection > encoding. We did not find a difference between baseline and Recollection. We apologize for the error and we corrected it in the main text.

With respect to the reviewer’s other remark: regions that went into this analysis were chosen based on the stationary functional connectivity results, as this was an analysis to better understand the interactions between these specific regions. We did ensure that our results were not spurious by applying a FDR-adjustment for multiple comparisons. Furthermore, we created matched AR1 randomized timeseries as a null-distribution for our comparisons, which is a more meaningful and valid control metric than choosing a region that may not interact with the MTL. These null-distributions are similar to the bootstrap and hence, the resampled timeseries have similar properties as the original data (distribution, autocorrelation) and have less bias in the rejection rates of the null-hypothesis. We have explained this further in the Materials and methods.

In the Results:

“We observed an increase in HIP-AMY coherence for encoding compared to baseline (β=-0.13, t(821)=-4.39, p<0.001, 95%CI[-0.20, -0.052]), a decrease in recollection compared to encoding (β=0.10, t(821)=3.37, p=0.004, 95%CI[0.023, 0.171]) and a trend for consolidation decrease compared to baseline (β=-0.07, t(821)=-2.35, p=0.088 95%CI[-0.141, 0.007]) and a change in phase from AMY leading the LC in encoding to the LC leading the AMY in recollection (β=0.91, t(821)=2.73, p=0.033, 95%CI[0.051,1.772] Figure 7B).”

In the Materials and methods:

“These null-distributions are similar to the bootstrap and hence, the resampled timeseries have similar properties as the original data (distribution, autocorrelation) and have less bias in the rejection rates of the null-hypotheses.”

[Editors' note: further revisions were suggested prior to acceptance, as described below.]

We thank the authors for implementing most of the requested changes. However, in their responses to Comment 1 and 2, they misunderstood or overlooked important aspects of the question. We would like to ask them to adjust their responses and changes in the paper as follows:Re Comment 1: Please also indicate whether the names used for the calculation of the bias score were paired with new or old faces or whether this was collapsed across both old and new faces.Re Comment 2: There was a misunderstanding in the response to comment 2. The statistics required in the table were the hit and false alarm rate for names, not for faces. Please report the Hit rate, FA rate and Hit-FA rate separately for the names paired in the memory tests with old emotional and old neutral faces and for those paired with new emotional and new neutral new faces.

We believe that there is misunderstanding in what these scores represent. The Recollection Hit Rate Raw represents the number of correct names to old faces identified during recollection divided by the total number of available old faces (whether or not they were identified as old). Thus, this is the %Hit rate. As indicated previously, 1 – Recollection Hit Rate (correction recognition) represents the false alarm. It should be noted that this %hit may be an underestimation as the participant may not have been able to indicate the name to the face, in case he/she identified the face as new. As explained in the previous revision and in Figure 1, only faces that were endorsed by the participant in the recognition stage would go to the recollection stage.

The %Hits as requested by the editor, would in our opinion be calculated as the number of correctly identified names belonging to old faces divided by the total number of presented old faces related to these names. This would then be expressed in % (multiplied by 100).

Therefore, the total number of old faces that were presented (denominator) equals the number of correctly recognized old faces in recognition. The correctly identified names (numerator) is the number of correct names to old faces during recollection.

The Recollection Hit Rate (correct Recognition) represents the number of correct names to old faces identified during recollection divided by the total number of old faces that were recognized correctly during recognition (presented as a %). This therefore represents the % Hit rate for names presented in the test with old faces that were correctly recognized as old and, to our understanding, is what the editor is requesting. 1 – Recollection Hit Rate (correct Recognition) represents the False Alarm for old faces.

We have added the Hit Rate – False Alarm rate to the table and the table is now included in the manuscript.

**References**

1. Khakpour-Taleghani B, Lashgari R, Aavani T, Haghparast A, Naderi N, Motamedi F. The locus coeruleus involves in consolidation and memory retrieval, but not in acquisition of inhibitory avoidance learning task. Behav Brain Res. 2008;189(2):257-62. doi: 10.1016/j.bbr.2008.01.004. PubMed PMID: 18295357.2. Wagatsuma A, Okuyama T, Sun C, Smith LM, Abe K, Tonegawa S. Locus coeruleus input to hippocampal CA3 drives single-trial learning of a novel context. Proc Natl Acad Sci U S A. 2018;115(2):E310-E6. doi: 10.1073/pnas.1714082115. PubMed PMID: 29279390; PubMed Central PMCID: PMCPMC5777050.3. Murchison CF, Zhang XY, Zhang WP, Ouyang M, Lee A, Thomas SA. A distinct role for norepinephrine in memory retrieval. Cell. 2004;117(1):131-43. Epub 2004/04/07. doi: 10.1016/s0092-8674(04)00259-4. PubMed PMID: 15066288.4. Hansen N, Manahan-Vaughan D. Locus Coeruleus Stimulation Facilitates Long-Term Depression in the Dentate Gyrus That Requires Activation of β-Adrenergic Receptors. Cereb Cortex. 2015;25(7):1889-96. doi: 10.1093/cercor/bht429. PubMed PMID: 24464942; PubMed Central PMCID: PMCPMC4459289.5. McIntyre CK, Hatfield T, McGaugh JL. Amygdala norepinephrine levels after training predict inhibitory avoidance retention performance in rats. Eur J Neurosci. 2002;16(7):1223-6. Epub 2002/10/31. doi: 10.1046/j.1460-9568.2002.02188.x. PubMed PMID: 12405982.6. Vazey EM, Moorman DE, Aston-Jones G. Phasic locus coeruleus activity regulates cortical encoding of salience information. Proc Natl Acad Sci U S A. 2018;115(40):E9439-E48. Epub 2018/09/21. doi: 10.1073/pnas.1803716115. PubMed PMID: 30232259; PubMed Central PMCID: PMCPMC6176602.7. Roozendaal B, Hermans EJ. Norepinephrine effects on the encoding and consolidation of emotional memory: improving synergy between animal and human studies. Curr Opin Behav Sci. 2017;14:115-22.8. Lemon N, Aydin-Abidin S, Funke K, Manahan-Vaughan D. Locus coeruleus activation facilitates memory encoding and induces hippocampal LTD that depends on β-adrenergic receptor activation. Cereb Cortex. 2009;19(12):2827-37. Epub 2009/05/14. doi: 10.1093/cercor/bhp065. PubMed PMID: 19435710; PubMed Central PMCID: PMCPMC2774396.9. Roozendaal B, Castello NA, Vedana G, Barsegyan A, McGaugh JL. Noradrenergic activation of the basolateral amygdala modulates consolidation of object recognition memory. Neurobiol Learn Mem. 2008;90(3):576-9. Epub 2008/07/29. doi: 10.1016/j.nlm.2008.06.010. PubMed PMID: 18657626; PubMed Central PMCID: PMCPMC2572617.10. Sara SJ, Devauges V. Priming stimulation of locus coeruleus facilitates memory retrieval in the rat. Brain Res. 1988;438(1-2):299-303. Epub 1988/01/12. doi: 10.1016/0006-8993(88)91351-0. PubMed PMID: 3345434.11. Sara SJ, Segal M. Plasticity of sensory responses of locus coeruleus neurons in the behaving rat: implications for cognition. Prog Brain Res. 1991;88:571-85. Epub 1991/01/01. doi: 10.1016/s0079-6123(08)63835-2. PubMed PMID: 1813935.12. Stanislaw H, Todorov N. Calculation of signal detection theory measures. Behavior Research Methods, Instruments, & Computers. 1999;31(1):137-49. doi: 10.3758/bf03207704.13. Yonelinas AP, Aly M, Wang WC, Koen JD. Recollection and familiarity: examining controversial assumptions and new directions. Hippocampus. 2010;20(11):1178-94. Epub 2010/09/18. doi: 10.1002/hipo.20864. PubMed PMID: 20848606; PubMed Central PMCID: PMCPMC4251874.14. Yonelinas AP, Otten LJ, Shaw KN, Rugg MD. Separating the brain regions involved in recollection and familiarity in recognition memory. J Neurosci. 2005;25(11):3002-8. Epub 2005/03/18. doi: 10.1523/JNEUROSCI.5295-04.2005. PubMed PMID: 15772360; PubMed Central PMCID: PMCPMC6725129.15. Eichenbaum H, Yonelinas AP, Ranganath C. The medial temporal lobe and recognition memory. Annu Rev Neurosci. 2007;30:123-52. Epub 2007/04/10. doi: 10.1146/annurev.neuro.30.051606.094328. PubMed PMID: 17417939; PubMed Central PMCID: PMCPMC2064941.16. Rugg MD, Vilberg KL. Brain networks underlying episodic memory retrieval. Current opinion in neurobiology. 2013;23(2):255-60. Epub 2012/12/05. doi: 10.1016/j.conb.2012.11.005. PubMed PMID: 23206590; PubMed Central PMCID: PMCPMC3594562.17. O'Brien KR, Kober T, Hagmann P, Maeder P, Marques J, Lazeyras F, Krueger G, Roche A. Robust T1-weighted structural brain imaging and morphometry at 7T using MP2RAGE. PLoS One. 2014;9(6):e99676. Epub 2014/06/17. doi: 10.1371/journal.pone.0099676. PubMed PMID: 24932514; PubMed Central PMCID: PMCPMC4059664.18. Avants BB, Tustison NJ, Song G, Cook PA, Klein A, Gee JC. A reproducible evaluation of ANTs similarity metric performance in brain image registration. NeuroImage. 2011;54(3):2033-44. Epub 2010/09/21. doi: 10.1016/j.neuroimage.2010.09.025. PubMed PMID: 20851191; PubMed Central PMCID: PMCPMC3065962.19. Klein A, Andersson J, Ardekani BA, Ashburner J, Avants B, Chiang M-C, Christensen GE, Collins DL, Gee J, Hellier P, Song JH, Jenkinson M, Lepage C, Rueckert D, Thompson P, Vercauteren T, Woods RP, Mann JJ, Parsey RV. Evaluation of 14 nonlinear deformation algorithms applied to human brain MRI registration. NeuroImage. 2009;46(3):786-802. doi: 10.1016/j.neuroimage.2008.12.037. PubMed PMID: PMC2747506.20. Iglesias JE, Augustinack JC, Nguyen K, Player CM, Player A, Wright M, Roy N, Frosch MP, McKee AC, Wald LL, Fischl B, Van Leemput K, Alzheimer's Disease Neuroimaging I. A computational atlas of the hippocampal formation using ex vivo, ultra-high resolution MRI: Application to adaptive segmentation of in vivo MRI. NeuroImage. 2015;115:117-37. doi: 10.1016/j.neuroimage.2015.04.042. PubMed PMID: 25936807; PubMed Central PMCID: PMC4461537.21. Iglesias JE, Van Leemput K, Bhatt P, Casillas C, Dutt S, Schuff N, Truran-Sacrey D, Boxer A, Fischl B, for the Alzheimer’s Disease Neuroimaging I. Bayesian segmentation of brainstem structures in MRI. NeuroImage. 2015;113:184-95. doi: 10.1016/j.neuroimage.2015.02.065. PubMed PMID: PMC4434226.22. Sclocco R, Beissner F, Desbordes G, Polimeni JR, Wald LL, Kettner NW, Kim J, Garcia RG, Renvall V, Bianchi AM, Cerutti S, Napadow V, Barbieri R. Neuroimaging brainstem circuitry supporting cardiovagal response to pain: a combined heart rate variability/ultrahigh-field (7 T) functional magnetic resonance imaging study. Philos Trans A Math Phys Eng Sci. 2016;374(2067). doi: 10.1098/rsta.2015.0189. PubMed PMID: 27044996; PubMed Central PMCID: PMCPMC4822448.23. Napadow V, Dhond R, Conti G, Makris N, Brown EN, Barbieri R. Brain correlates of autonomic modulation: combining heart rate variability with fMRI. NeuroImage. 2008;42(1):169-77. Epub 2008/06/06. doi: 10.1016/j.neuroimage.2008.04.238. PubMed PMID: 18524629; PubMed Central PMCID: PMCPMC2603289.24. Lane RD, McRae K, Reiman EM, Chen K, Ahern GL, Thayer JF. Neural correlates of heart rate variability during emotion. NeuroImage. 2009;44(1):213-22. Epub 2008/09/10. doi: 10.1016/j.neuroimage.2008.07.056. PubMed PMID: 18778779.25. Thayer JF, Ahs F, Fredrikson M, Sollers JJ, 3rd, Wager TD. A meta-analysis of heart rate variability and neuroimaging studies: implications for heart rate variability as a marker of stress and health. Neurosci Biobehav Rev. 2012;36(2):747-56. Epub 2011/12/20. doi: 10.1016/j.neubiorev.2011.11.009. PubMed PMID: 22178086.26. Bittiner SB, Smith SE. Beta-adrenoceptor antagonists increase sinus arrhythmia, a vagotonic effect. Br J Clin Pharmacol. 1986;22(6):691-5. Epub 1986/12/01. doi: 10.1111/j.1365-2125.1986.tb02959.x. PubMed PMID: 2882771; PubMed Central PMCID: PMCPMC1401218.27. Chang C, Glover GH. Time–frequency dynamics of resting-state brain connectivity measured with fMRI. NeuroImage. 2010;50(1):81-98. doi: https://doi.org/10.1016/j.neuroimage.2009.12.011.

28. Rohleder N, Nater UM. Determinants of salivary α-amylase in humans and methodological considerations. Psychoneuroendocrinology. 2009;34(4):469-85. doi: 10.1016/j.psyneuen.2008.12.004. PubMed PMID: 19155141.

29

. Jacobs HIL, Riphagen JM, Razat CM, Wiese S, Sack AT. Transcutaneous vagus nerve stimulation boosts associative memory in older individuals. Neurobiology of Aging. 2015;36(5):1860-7. doi: 10.1016/j.neurobiolaging.2015.02.023. PubMed PMID: WOS:000355100900008.30. Akselrod S, Gordon D, Ubel FA, Shannon DC, Berger AC, Cohen RJ. Power spectrum analysis of heart rate fluctuation: a quantitative probe of beat-to-beat cardiovascular control. Science. 1981;213(4504):220-2. Epub 1981/07/10. doi: 10.1126/science.6166045. PubMed PMID: 6166045.31. Wallentin M, Nielsen AH, Vuust P, Dohn A, Roepstorff A, Lund TE. Amygdala and heart rate variability responses from listening to emotionally intense parts of a story. NeuroImage. 2011;58(3):963-73. Epub 2011/07/14. doi: 10.1016/j.neuroimage.2011.06.077. PubMed PMID: 21749924.32. Billings JCW, Thompson GJ, Pan WJ, Magnuson ME, Medda A, Keilholz S. Disentangling Multispectral Functional Connectivity With Wavelets. Front Neurosci. 2018;12:812. Epub 2018/11/22. doi: 10.3389/fnins.2018.00812. PubMed PMID: 30459548; PubMed Central PMCID: PMCPMC6232345.33. Chang C, Leopold DA, Scholvinck ML, Mandelkow H, Picchioni D, Liu X, Ye FQ, Turchi JN, Duyn JH. Tracking brain arousal fluctuations with fMRI. Proc Natl Acad Sci U S A. 2016;113(16):4518-23. Epub 2016/04/07. doi: 10.1073/pnas.1520613113. PubMed PMID: 27051064; PubMed Central PMCID: PMCPMC4843437.34. Chang C, Liu Z, Chen MC, Liu X, Duyn JH. EEG correlates of time-varying BOLD functional connectivity. NeuroImage. 2013;72:227-36. Epub 2013/02/05. doi: 10.1016/j.neuroimage.2013.01.049. PubMed PMID: 23376790; PubMed Central PMCID: PMCPMC3602157.35. Fukunaga M, Horovitz SG, van Gelderen P, de Zwart JA, Jansma JM, Ikonomidou VN, Chu R, Deckers RH, Leopold DA, Duyn JH. Large-amplitude, spatially correlated fluctuations in BOLD fMRI signals during extended rest and early sleep stages. Magn Reson Imaging. 2006;24(8):979-92. Epub 2006/09/26. doi: 10.1016/j.mri.2006.04.018. PubMed PMID: 16997067.36. Horovitz SG, Fukunaga M, de Zwart JA, van Gelderen P, Fulton SC, Balkin TJ, Duyn JH. Low frequency BOLD fluctuations during resting wakefulness and light sleep: a simultaneous EEG-fMRI study. Hum Brain Mapp. 2008;29(6):671-82. Epub 2007/06/29. doi: 10.1002/hbm.20428. PubMed PMID: 17598166.37. Olbrich S, Mulert C, Karch S, Trenner M, Leicht G, Pogarell O, Hegerl U. EEG-vigilance and BOLD effect during simultaneous EEG/fMRI measurement. NeuroImage. 2009;45(2):319-32. Epub 2008/12/27. doi: 10.1016/j.neuroimage.2008.11.014. PubMed PMID: 19110062.38. De Havas JA, Parimal S, Soon CS, Chee MW. Sleep deprivation reduces default mode network connectivity and anti-correlation during rest and task performance. NeuroImage. 2012;59(2):1745-51. Epub 2011/08/30. doi: 10.1016/j.neuroimage.2011.08.026. PubMed PMID: 21872664.39. Wong CW, Olafsson V, Tal O, Liu TT. The amplitude of the resting-state fMRI global signal is related to EEG vigilance measures. NeuroImage. 2013;83:983-90. Epub 2013/08/01. doi: 10.1016/j.neuroimage.2013.07.057. PubMed PMID: 23899724; PubMed Central PMCID: PMCPMC3815994.40. Lurie DJ, Kessler D, Bassett DS, Betzel RF, Breakspear M, Kheilholz S, Kucyi A, Liegeois R, Lindquist MA, McIntosh AR, Poldrack RA, Shine JM, Thompson WH, Bielczyk NZ, Douw L, Kraft D, Miller RL, Muthuraman M, Pasquini L, Razi A, Vidaurre D, Xie H, Calhoun VD. Questions and controversies in the study of time-varying functional connectivity in resting fMRI. Netw Neurosci. 2020;4(1):30-69. Epub 2020/02/12. doi: 10.1162/netn_a_00116. PubMed PMID: 32043043; PubMed Central PMCID: PMCPMC7006871.41. Shine JM, van den Brink RL, Hernaus D, Nieuwenhuis S, Poldrack RA. Catecholaminergic manipulation alters dynamic network topology across cognitive states. Netw Neurosci. 2018;2(3):381-96. Epub 2018/10/09. doi: 10.1162/netn_a_00042. PubMed PMID: 30294705; PubMed Central PMCID: PMCPMC6145851.